# Ozone trends over the United States at different times of day

Yingying Yan[1], Jintai Lin[1], Cenlin He[2]

[1] Laboratory for Climate and Ocean-Atmosphere Studies, Department of Atmospheric and Oceanic Sciences, School of Physics, Peking University, Beijing 100871, China

[2] Joint Institute for Regional Earth System Science and Engineering and Department of Atmospheric and Oceanic Sciences, University of California, Los Angeles, Los Angeles, California, USA

Email: linjt@pku.edu.cn

## Abstract

In the United States, the decline of summertime daytime peak ozone in the last 20 years has been clearly connected to reductions in anthropogenic emissions. Yet questions remain on how and through what mechanisms ozone at other times of day have changed over the recent decades. Here we analyze the interannual variability and trends of ozone at different hours of day, using observations from about 1000 US sites during 1990–2014. We find a clear diurnal cycle both in the magnitude of ozone trends and in the relative importance of climate variability versus anthropogenic emissions to ozone changes. Interannual climate variability has mainly been associated with the de-trended fluctuation in the US annual daytime ozone over 1990–2014, with a much smaller effect on the nighttime ozone. Reductions in anthropogenic emissions of nitrogen oxides have led to substantial growth in the US annual average nighttime ozone due to reduced ozone titration, while the summertime daytime ozone has declined. Environmental policymaking might consider further improvements to reduce ozone levels at night and other non-peak hours.

## 1. Introduction

Tropospheric ozone is a potent pollutant damaging human and ecological health. The United States Environmental Protection Agency (EPA) targets the maximum daily 8-hour average (MDA8) ozone levels for regulation, with a current standard at 70 parts per billion (ppb). The Global Burden of Disease assessment (Brauer et al., 2016;Lim et al., 2012;Forouzanfar et al., 2015;Forouzanfar et al., 2016), however, estimates the threshold level (below which exposure to ozone is not harmful) to be between 33.3 ug/m$^3$ and 41.9 ug/m$^3$ (between 15.5 ppb and 19.5 ppb). Further, additional epidemiological evidence has shown that there is not a real threshold, and ozone has adverse health effects at all concentrations (Bell et al., 2006;Peng et al., 2013;Yang et al., 2012).

Chemically, surface ozone is produced in the sunshining daytime and destroyed mainly by nitrogen oxides ($NO_x$) at night, with a transition between production and destruction in the dawn and dusk hours. Understanding of ozone changes and drivers at different times of day might provide additional information to assist further ozone mitigation policymaking beyond the MDA8

regulation. Previous observational and modeling studies have revealed important impacts of varying climate conditions and anthropogenic precursor emissions on the near-surface daytime, MDA8 or daily mean ozone over the United States (US) (Jacob and Winner, 2009; Fiore et al., 2015). A particular finding for these studies is that the US emission reductions have decreased the summertime daytime peak ozone over much of the United States (Cooper et al., 2012;Lefohn et al., 2010;Simon et al., 2015;Strode et al., 2015).

In addition, most studies on the US ozone trends/variability tend to focus on ozone changes in a particular season and/or over a particular region – for example, summertime ozone (Rieder et al., 2015;Lu et al., 2016; Reddy and Pfister, 2016) over the eastern US (Zhang and Wang, 2016; Rieder et al., 2015; Shen et al., 2015) or springtime ozone (Cooper et al., 2010) over the western US (Lin et al., 2015). Lin et al. (2017) analyzed the various driving factors of the US MDA8 ozone trends by season and region, with no discussion on ozone at other hours.

Bloomer et al. (2010) analyzed the 1989–2007 changes in the diurnal cycle of ozone observed from five stations over the eastern US. They showed ozone reductions at most times of day in the warm seasons due to emission reductions, in contrast to the increases in winter. Jhun et al. (2015) used a statistical model of ozone, nitrogen oxides and several meteorological parameters to analyze the ozone trends over 1994–2010 measured at over 100 sites across the US. They linked the observed reductions in nitrogen oxides to reductions in warm season peak ozone and to enhancements in cold season peak ozone and warm season nighttime ozone. Overall, the historical changes of ozone at night and other non-peak hours and their underlying climatic or emission causes have been much less studied compared to those for peak ozone.

Here, with the usage of hourly data observed at about 1000 sites from the Air Quality System (AQS) network, we contrasted the interannual variations of the daytime versus the nighttime US ozone over 1990–2014, and also estimated the trends for ozone at the different hours of the day. We further quantified the individual effects of interannual climate variability and anthropogenic emissions on the ozone change, by using three climate indices and simulations of the chemical transport model (CTM) GEOS-Chem. We focused on the large-scale features of annual mean US averaged surface ozone and the impacts of broad changes in emissions and climate. We also contrasted the ozone changes and drivers between the eastern and western US and between different seasons, with complementary discussions for individual locations and land use types (urban, suburban and rural).

The manuscript is structured as follows. Section 2 introduces the observation data, climate indices, and GEOS-Chem simulations. Section 3 analyzes the linear trends for annual mean hourly, daytime mean, nighttime mean and daily mean ozone. Section 4 compares the observed ozone trends and interannual variability with three climate indices relevant to the US air quality. Section 5 uses four GEOS-Chem simulations, with perturbed emissions and meteorological inputs, to quantify the individual effects of climate variability and anthropogenic emissions. Section 6 concludes the study.

## 2. Data and Methods

## 2.1 Ozone measurements

Hourly measurements of ground-level ozone over 1990–2014 are taken from about 1000 AQS sites (http://aqsdr1.epa.gov/aqsweb/aqstmp/airdata/download_files.html). For a given year, the number of measurement sites vary from 825 to 1295 (see Fig. 1 for site distribution). The fraction of hours in any year with missing data ranges from 21.3% to 28.5%. As indicated in Fig. 1, the AQS network includes rural (42% of sites on average), suburban (40%) and urban (18%) sites, based on the official site description document. We mapped the ozone measurements on a $2.5^o$ long. x $2^o$ lat. grid to facilitate a comparison with GEOS-Chem simulations. For each hour, we averaged all available data in a given grid cell. In order to contrast the daytime and nighttime ozone changes and their underlying drivers, we calculated daily mean, daytime (07:00–19:00 local time) mean, and nighttime (19:00–07:00 local time) mean ozone mixing ratios from the gridded hourly data. We averaged the daily data to produce monthly mean and then annual mean values. We finally selected a total of 124 grid cells with annual average values available in all years.

*Robustness of data selection method*

To test the robustness of ozone trends results against the data/site selection method, we used four alternate methods to choose sites, as follows.

The first alternate choice concerns the number of sites in a particular grid cell for geographical representativeness, and it excludes 24 out of the default 124 grid cells that cover less than three sites each.

The second choice concerns the temporal continuity of valid data at each site, and it only includes sites with valid data in at least three years for every five years (1990–1994, 1995–1999, etc.) – this leads to a total of 94 valid grid cells that cover 131 rural sites, 164 suburban sites, and 102 urban sites. The spatial distribution of ozone trends in these 94 grid cells is consistent with the trends in the default 124 grid cells (not shown), although the (number and locations of) sites in each common grid cell differ between the two site selection methods.

The third choice is much stricter, and it only selects 70 sites (24 rural sites, 27 suburban sites, and 19 urban sites) with valid hourly data in 75% or more of hours during 1990–2014.

The last choice is more complex, and is similar to the method adopted by Cooper et al (Cooper et al., 2012). At a given site, if more than 50% of hourly data are missing in any daytime or nighttime, then the particular day is discarded. If more than 50% of days in any season do not contain valid data, then the particular season is discarded. For any season, there must be valid seasonal mean data in at least 20 out of 25 years during 1990–2014, otherwise data in all years for the particular season are discarded. These criteria lead to 82 sites with valid data, including 30

rural sites, 34 suburban sites, and 18 urban sites.

Table 1 shows that our default data selection method lead to ozone trends similar to the four
alternate methods, for the US mean ozone on an annual basis. Across the five methods, the
growth rates are about 0.14–0.17 ppb/yr for the US annual mean daily mean ozone, 0.06–0.09
5    ppb/yr for the daytime mean ozone, and 0.21–0.24 ppb/yr for the nighttime mean ozone.
Furthermore, the interannual variation of US annual ozone in our default case is highly correlated
to the other four cases, no matter whether the time series are de-trended (R = 0.84–0.95). We thus
conclude that our ozone trends and variability results are robust against our choice of sites/data.

**2.2 Climate indices**

We relate the interannual variability of ozone to two major climate indices relevant to the US air
quality (Sutton et al., 2007; Lin et al., 2015): the Atlantic Multi-decadal Oscillation (AMO;
https://www.esrl.noaa.gov/psd/data/timeseries/AMO/) index and the Oceanic Niño index (ONI;
http://origin.cpc.ncep.noaa.gov/products/analysis_monitoring/ensostuff/ONI_v5.php).

De-trended annual AMO index time series over 1990–2014 is calculated from the unsmoothed
Kaplan sea surface temperature (SST) dataset of the National Oceanic and Atmospheric
Administration        (NOAA)        Earth        System        Research        Laboratory
(http://www.esrl.noaa.gov/psd/data/correlation/amon.us.data). Observational and model studies
have shown that the recent multi-decadal fluctuations in Atlantic SST were associated with
large-scale climate anomalies (Enfield et al., 2001; Johannessen et al., 2004; Hu et al., 2011;
Oglesby et al., 2012) important for the ozone chemistry. The warm conditions in the North
Atlantic Ocean (positive AMO) have been associated with increased air temperatures at northern
latitudes (by more than 3°C locally especially in winter) (Johannessen et al., 2004) and reduced
summertime rainfall and increased droughts over much of the US (Enfield et al., 2001; McCabe
et al., 2004). Delworth and Mann (2000) showed evidence of AMO-related variations in sea level
pressure (SLP), and hence atmospheric circulation, in the North Atlantic region. Such
AMO-associated meteorological changes can alter the distribution of tropospheric constituents
(Olsen et al., 2016), including ozone (Shen et al., 2015; Lin et al., 2014). For example, warmer
temperature under a positive AMO phase tends to enhance the ozone production over the US (Lin
et al., 2014); less precipitation means less cloudy and higher radiation for photochemistry
(Kunkel et al., 2008); and the change in circulation affects the ozone transport (Fu et al., 2015).

De-trended annual ONI index time series over 1990–2014 is calculated from the NOAA Climate
Prediction                          Center                          dataset
(http://www.cpc.noaa.gov/products/analysis_monitoring/ensostuff/ensoyears.shtml). The ONI
index refers to ERSST.v4 SST anomalies in the Niño 3.4 region, as an indicator of the El
Niño-Southern Oscillation (ENSO). These SST anomalies are associated with global climate
variability, including changes in the temperature and precipitation patterns over the Unites States
(Ropelewski and Halpert, 1987; Yu et al., 2012; Yu and Zou, 2013; Liang et al., 2015), through

alterations in atmospheric circulations and teleconnection patterns (Bjerknes, 1969; Enfield, 1989). The Niño 3.4 index (referred as ONI here) has been widely used to examine the impact of ENSO on surface ozone over the United States (e.g., Lin et al., 2015; Xu et al., 2017). Xu et al. (2017) showed that over 1993–2013, the monthly ozone decreases (increases) during El Niño (La Niña) years, with the amplitude varying from 0.4 ppb (for the US average) up to 1.8 ppb (for the southeastern US) per standard deviation of the Niño 3.4 index. The La Niña years (strongly negative ONI) tend to be associated with a more meandering jet over the central western US in favor of stratosphere-troposphere exchange (STE) of ozone but not transport from Asia (Lin et al., 2015).

In order to indicate the climate variability that influence the whole US, we combined the de-trended and normalized AMO and ONI indices to obtain a third index, named AMONI: $AMONI = AMO_{detrended_{normalized}} - ONI_{detrended_{normalized}}$. The normalization could adjust the AMO and ONI values measured on different scales to a common scale and keep the individual characteristics of the original AMO and ONI indices. The negative sign for ONI in the formula accounts for the negative correlation between de-trended ozone and ONI anomalies (see Sect. 4). Thus a positive AMO and a negative ONI, both of which are closely related to higher ozone mixing ratios over the US, contribute to a positive AMONI index.

A recent work by Shen et al. (2017) developed two metrics, MAM-ΔSST and MAM-ΔSLP, to study June-July-August (JJA) MDA8 ozone variability across much of the eastern US. They found that MAM-ΔSST is highly correlated to the summer ozone (R = ~ 0.7), which level of correlation is comparable to our results for daytime and daily mean ozone (see Sect. 4).

## 2.3 Model simulations

We used the global chemical transport model GEOS-Chem (version 9-02, http://wiki.seas.harvard.edu/geos-chem/index.php/Main_Page) to simulate the US surface ozone changes over 2004–2012. GEOS-Chem has been used extensively for ozone studies (e.g., Shen et al., 2015;Fu et al., 2015;Yan et al., 2016;Zhang and Wang, 2016). Here, the model is run at a horizontal resolution of 2.5 °long. x 2 °lat. with 47 vertical layers (including 10 layers of ~ 130 m thickness each below 850 hPa), as driven by the GEOS-5 assimilated meteorological fields. The model is run with the standard $HO_x-NO_x-VOC$-ozone-aerosol chemistry (Mao et al., 2013) with some recent updates (Yan et al., 2014). The Linoz scheme is used for the stratospheric ozone production (McLinden et al., 2000). Vertical mixing in the planetary boundary layer employs a non-local scheme implemented by Lin and McElroy (2010). Model convection adopts the Relaxed Arakawa-Schubert scheme (Rienecker et al., 2008).

A "Control" simulation includes variations in meteorology and anthropogenic emissions, and three sensitivity simulations keep meteorology or anthropogenic emissions constant throughout the years. As the GEOS-5 meteorological fields are only available from December 2003 to March 2013, all GEOS-Chem simulations are from December 2003 through December 2012, with

results analyzed for 2004–2012.

*Emissions in the "Control" simulation of GEOS-Chem*

Global and regional anthropogenic emission inventories used here are summarized in Yan et al. (2016). Global anthropogenic emissions for CO and $NO_x$ from 2004 to 2008 are taken from the Emission Database for Global Atmospheric Research (EDGAR) v4.2 inventory. Global anthropogenic emissions of NMVOC use the REanalysis of the TROpospheric chemical composition (RETRO) monthly global inventory for 2000 (Hu et al., 2015). Emissions over China, Asia, the US, Mexico, Canada and Europe are further replaced by the MEIC (base year is 2008; www.meicmodel.org), INTEX-B (base year is 2006 (Zhang et al., 2009)), NEI05 (base year is 2005, ftp://aftp.fsl.noaa.gov/divisions/taq/), BRAVO (base year is 1999 (Kuhns et al., 2003)), CAC (base year is 2005, http://www.ec.gc.ca/pdb/cac/cac_home_e.cfm), and EMEP (base year is 2005 (Auvray and Bey, 2005)) regional inventories, respectively. Emission data include monthly or seasonal variability (Yan et al., 2016).

Most anthropogenic emission inventories provide data for a base year. In order to simulate the interannual variability of ozone, we scaled $NO_x$ and CO emissions from the base year to other years between 2004 and 2012. Over the US, China and Canada, emissions of $NO_x$ are scaled based on the tropospheric $NO_2$ columns from OMI measurements (Lin et al., 2015;Vinken et al., 2014). For the US (for CO), Canada (for CO), and Europe (for $NO_x$ and CO), emissions are scaled according to NEI (http://www3.epa.gov/ttn/chief/trends/index.html), Environment Canada National Pollutant Release Inventory Trends (http://www.ec.gc.ca/inrp-npri/), and European Monitoring and Evaluation Program (http://www.emep.int/). For regions not affected by the above scaling processes, $NO_x$ and CO emissions are scaled according to EDGAR (for 2004–2008), or to changes in total and liquid fuel $CO_2$ emissions, respectively, following van Donkelaar et al. (van Donkelaar et al., 2008) (for 2009–2012). $CO_2$ emissions are taken from the Carbon Dioxide Information Analysis Center (http://cdiac.ornl.gov/).

Monthly biomass burning emissions are taken from the Global Fire Emissions Database version 3 (GFED3) (van der Werf et al., 2010). Other natural emissions (lightning $NO_x$, soil $NO_x$, and biogenic NMVOC) are parameterized based on model meteorology. Lightning $NO_x$ emissions are parameterized based on cloud top heights (Price and Rind, 1992), and are further constrained by the lightning flash counts detected from the satellite instruments (Murray et al., 2012;Murray et al., 2013). Soil $NO_x$ emissions follow Hudman et al. (2012). Biogenic emissions of NMVOC follow the Model of Emissions of Gases and Aerosols from Nature (MEGAN v2.1) with the Hybrid algorithm (Guenther, 2007;Guenther et al., 2012).

Figure 2 shows monthly anthropogenic and natural emissions of CO, $NO_x$, and NMVOC over the United States from 2004 to 2012, as used in the "Control" simulation. Averaged over 2004–2012, the US emissions (from all sources) are about 69.4 Tg/yr for CO, 6.6 TgN/yr for $NO_x$, and 34.0 TgC/yr for NMVOC. Anthropogenic emissions are the dominant source for CO, are comparable

to natural sources for NO$_x$, and are a minor source for NMVOC. Anthropogenic emissions of NO$_x$ and CO decline rapidly at rates of 0.25 TgN/yr (4.1%/yr relative to 2004) for NO$_x$ and 2.7 Tg/yr (3.2%/yr) for CO. Natural emissions vary from one year to another with no obvious trends.

The "Control" simulation accounts the interannual variations in climate and anthropogenic emissions of NOx and CO. Between 2004 and 2012, anthropogenic emissions of NOx and CO in the US decline by 33% and 26%, respectively (Fig. 2a,b). As the US anthropogenic emissions of NMVOC are smaller than natural emissions by a factor of about 7 (Fig. 2c), their reduction from 2004 to 2012 (by ~ 9%) is not included here. A sensitivity simulation suggests that including changes in anthropogenic NMVOC emissions result in ozone changes from 2004 to 2012 very close to the "Control" simulation (see Sect. 5.1).

**3. Observed ozone trends at different times of day**

The black line in Fig. 3 shows the 1990–2014 US average ozone trends at the individual hours of the day (local standard time). At nighttime hours, the US annual mean ozone grows relatively constantly at a statistically significant rate of about 0.2 ppb/yr. The growth rate declines in the morning hours and increases during the late afternoon hours. The minimum growth rate is located at around 14:00, when the ozone level peaks, and is slightly negative (but insignificant) with a value of -0.01$\pm$0.16 ppb/yr. The general characteristics of ozone trends are consistent with the results of Jhun et al. (2015). The ozone trends for individual hours tend to weaken the diurnal cycle of ozone – in particular, the diurnal range of ozone (i.e., maximum – minimum) is reduced by 15% from 26.4 ppb in 1990–1994 to 22.4 ppb in 2010–2014. The contrasting ozone trends between the daytime and the nighttime are indicative of distinctive causes. Hereafter we will focus on trends and (de-trended) variability in the daytime mean, nighttime mean and daily mean ozone, unless stated otherwise.

Table 2 shows the 1990–2014 trends for the US annual average daytime, nighttime and daily mean ozone, and Fig. 4a shows their time series. The US annual average daily mean ozone grows at a rate of 0.16 ppb/yr (P-value < 0.01 according to an F-test). The growth mainly reflects enhanced nighttime mean ozone (at 0.21 ppb/yr, P-value < 0.01). The daytime mean ozone grows at a much lower rate of 0.09 ppb/yr (P-value < 0.05). The implied total growth from 1990 to 2014 is about 4.1 ppb, 2.3 ppb and 5.3 ppb for daily mean, daytime and nighttime ozone, respectively. Similar to the enhanced US annual average ozone, increasing trends of ~1 ppb/yr in the annual mean ozone are observed at mountainous sites (e.g., Tanimoto et al., 2009) and regional background stations (e.g., Wang et al., 2009) in Asia. In contrast, European annual mean ozone levels have on average been decreasing during the last 20 years (e.g., Sicard et al, 2013). Furthermore, annual mean surface ozone at a background station in eastern China has declined (Xu et al., 2008).

Table 2 also differentiates the ozone trends for individual seasons over the eastern and western United States (separated by 100 ˚W). Overall, the growth rates are higher over the west than over the east, and the regional difference reaches about 0.15 ppb/yr in summer (June, July and August)

for daytime, nighttime and daily mean ozone. Seasonally, the most significant growth occurs in spring, with growth rates at 0.17–0.26 ppb/yr for the US average daytime, nighttime and daily mean ozone. For the nighttime ozone, the range of growth rates across the seasons is smaller over the west (0.17–0.30 ppb/yr) than over the east (0.05–0.24 ppb/yr). For the daytime ozone over the east, the increases in spring, fall and winter (0.07–0.15 ppb/yr) are contrasted by a reduction in summer (-0.11 ppb/yr), resulting in a weakened trend for the annual mean ozone (0.06 ppb/yr). The decreases in summertime ozone are associated with reductions in the US anthropogenic emissions of precursor gases $NO_x$, carbon monoxide (CO), and non-methane volatile organic compounds (NMVOC) revealed from the National Emissions Inventory (Fig. 4b), reflecting the success in controlling peak-level ozone (Cooper et al., 2012;Lefohn et al., 2010;Simon et al., 2015). Similarly, a substantial decrease in precursor concentrations over the last decades in Europe, in line with the long-term emission declines (Colette et al., 2011; Wilson et al., 2012), has resulted in a reduction in ozone episodes (Guerreiro et al., 2014). In contrast, the warm season afternoon ozone over eastern China has been growing at rates of 1–3 ppb/yr over the past 20 years (Sun et al., 2016; and references therein) as a result of rapidly growing precursor emissions.

Figure 5 further shows the spatial distribution of ozone trends across the 124 grid cells for daily mean, nighttime mean and daytime mean ozone in individual seasons. The nighttime ozone has grown or remained constant in most seasons and grid cells, although ozone reductions are also visible in the summertime eastern US and at sparse places in other seasons. For the daytime ozone, the summertime declines over the eastern United States and southern California are contrasted by the summertime increases at other places and the springtime increases over most regions. The springtime daytime increases over the western US are consistent with the MDA8 ozone growth rates (0.2–0.5 ppb/yr) reported by Lin et al. (2017). Averaged across all seasons, the nighttime ozone has grown in most grid cells, while the daytime ozone has grown over the west with mixed trends over the east (Fig. 5b, c). The spatial distribution of the trends in the annual daily mean ozone (Fig. 5a) much resembles the trends in the nighttime ozone. The local patterns in ozone trends reflect the effects of small-scale emission, chemical and/or meteorological features (Cooper et al., 2012;Jhun et al., 2015;Simon et al., 2015;Strode et al., 2015).

Table 3 presents the trends at the 5th, 50th and 95th percentiles for the daytime mean, nighttime mean and daily mean ozone, separately for urban, suburban and rural sites. For each site, we calculated daily mean, daytime mean, and nighttime mean ozone mixing ratios in each day, and then computed the 5th, 50th, and 95th ozone percentiles in each year. We then averaged the annual data over each site type (rural, suburban, or urban) for a subsequent trend analysis. Overall, the 50th percentile trends are positive for daytime, nighttime and daily mean ozone at all site types, consistent with the trends in the US annual mean ozone (Table 2).

Table 3 shows that for the nighttime mean ozone, trends at the 5th, 50th and 95th percentiles are relatively consistent: the 5th percentile ozone grows by 0.19–0.21 ppb/yr across the three site

types, the 50th percentile ozone grows by 0.19–0.22 ppb/yr, and the 95th percentile ozone increases at 0.22–0.27 ppb/yr.

Table 3 further shows that for the daytime mean ozone, trends at the 5th and 95th percentiles differ greatly, and there are only small differences across the rural, suburban and urban sites. The 5th percentile daytime mean ozone grows rapidly by 0.28–0.29 ppb/yr across the three site types, while the 95th percentile ozone declines by 0.13–0.21 ppb/yr. At the rural sites, the growth rate is about 0.29 ppb/yr (P-value < 0.01) at the 5th percentile and -0.13 ppb/yr (P-value < 0.01) at the 95th percentile. For the 50th percentile daytime mean ozone, trends differ greatly between rural, suburban and urban sites: the growth rate varies from 0.02 ppb/yr (statistically insignificant, P-value = 0.95) for the rural sites to 0.19 ppb/yr (P-value < 0.01) for the urban sites.

Overall, the distinctive trends at the three percentiles reflect a decrease in the peak ozone contrasted by an increase in the low ozone (Table 3), a decrease in summer compensated by an increase in other seasons (Table 2), and stronger tendency of ozone growth over the west than over the east (Table 2 and Fig. 5).

Our daytime results in Table 3 are consistent with Cooper et al. who examined the afternoon ozone trends over 1990–2010 at 53 rural sites (Cooper et al., 2012). Our 5th and 95th percentile daytime ozone trends at rural, suburban and urban sites are also broadly consistent with Simon et al. who analyzed the trends over 1998–2013 in the MDA8 ozone (Simon et al., 2015). The summertime decreases, largest at the 95th percentile, and wintertime increases in the 50th to 5th percentiles over the eastern US are also showed in the MDA8 ozone results of Lin et al. (2017) for 70 rural sites. Note that these previous studies (Cooper et al., 2012; Simon et al., 2015; Lin et al., 2017) were focused on the high-ozone afternoon hours and excluded the low-ozone early morning hours that have experienced notable ozone growth (black line in Fig. 3).

**4. Relationship between de-trended ozone and climate variability**

We relate the interannual variability of ozone to AMONI, AMO and ONI. We de-trended the time series of ozone and climate indices with linear fit for a subsequent statistical analysis; de-trending the ozone time series removed the effects of continuously rising Asian emissions and declining US emissions (Zhang et al., 2008; Lin et al., 2008; Huang et al., 2008; Cooper et al., 2010; Simon et al., 2015; Jhun et al., 2015; Verstraeten et al., 2016; Lin et al., 2017).

Figure 6a-c shows the time series of de-trended annual AMONI, AMO, and ONI indices between 1990 and 2014, in comparison with the de-trended ozone data. The three indices are interannually consistent with the daytime and daily mean ozone anomalies. The AMONI anomaly correlates strongly to the daytime (R = 0.71, P-value < 0.01 by a one-sided T-test) and daily mean (R = 0.62, P-value < 0.01) ozone anomalies, stronger than AMO and ONI. The negative AMONI anomalies in the early 1990s, 2009 and 2014, contributed by negative AMO and positive ONI (El Niño-like), correspond to negative anomalies in the daytime and daily mean ozone. By comparison, the positive AMONI anomalies around the late 1990s and early 2000s are associated with positive

ozone anomalies.

Figure 7a, b further shows the correlation between de-trended AMONI and de-trended daily and daytime mean ozone in individual grid cells. The AMONI-ozone correlation is positive and statistically significant over most of the eastern US, and it reaches 0.82 over parts of the southeast. A positive AMONI anomaly is associated with increased temperature over the east (Fig. 8c), which enhances biogenic emissions, changes chemical reaction rates, and changes atmospheric circulation that overall lead to increased ozone formation/buildup (Jacob and Winner, 2009; Shen et al., 2015; Fu et al., 2015; Xu et al., 2017). AMONI also correlate positively to ozone over the high-altitude west. This is because a negative ONI anomaly (La Niña-like) means a decrease in lower-tropospheric transport of ozone-poor air from the Eastern Pacific (Xu et al., 2017) and a more meandering subtropical jet and strengthened ozone transport from the stratosphere that compensates for weakened transport from Asia (Lin et al., 2015). AMONI correlates negatively to ozone over southern California, likely reflecting reduced temperature there associated with a positive AMONI (negative ONI) anomaly (Fig. 8b, c).

Figure 6a-c shows that the de-trended annual nighttime ozone anomaly corresponds weakly to the three climate indices, with statistically insignificant correlations at 0.15–0.34 (P-value = 0.39–0.78). Figure 7 also shows that AMONI is statistically significantly correlated to the de-trended nighttime ozone only in a few grid cells. We also found statistically insignificant de-trended correlations between the nighttime ozone and other climate indices such as the Pacific Decadal Oscillation (R = 0.06), the Arctic Oscillation (R = -0.05), and the North Atlantic Oscillation (R = -0.29). A possible reason for weaker correlations between climate indices and the nighttime ozone anomaly (compared to the daytime and daily anomalies) is the distinct chemistry at night (i.e., titration by nitrogen), compared to the daytime photochemistry. As the vertical mixing weakens at night, the chemical process becomes more localized and may be less sensitive to large-scale climate variability.

Table 4 further shows the seasonal and regional differences in the correlation between de-trended AMONI and de-trended ozone. Over the eastern US, the correlations for daytime ozone reach 0.72 in summer and 0.74 in fall. The correlations for nighttime ozone are also relatively large (0.55–0.60) due to influences by "residual" ozone transitioned from the daytime (see Sect. 5.2 for diurnal cycle of model-observation correlations). The correlations are very weak (0.06–0.18) in winter and spring for both daytime and nighttime ozone. Over the western US, the correlations do not exceed 0.50 in all seasons for daytime, nighttime and daily mean ozone, smaller than those over the eastern US. This is likely due to compensating effects on different transport and chemical processes. For example, a positive AMONI are associated with enhanced ozone production (due to enhanced temperature and reduced precipitation in the context of a positive AMO), weakened trans-pacific ozone transport from Asia (due to a negative ONI with a La Niña pattern), and strengthened STE. Nonetheless, the correlations for the western US are higher in winter and spring (0.37–0.50) than in summer and fall (0.19–0.27). Overall, Table 4 and Fig. 7 suggest that the AMONI-associated large-scale climate variability (AMO and ENSO) affects the

warm season eastern US ozone (likely via chemical processes) and cold season western US ozone (likely via dynamic transport processes).

## 5. Effects of emissions and climate variability on ozone revealed by GEOS-Chem simulations

We further used GEOS-Chem simulations to investigate the distinctive effects of interannual climate variability and anthropogenic emissions on the US daytime and nighttime ozone.

We investigated the simulated US ozone changes and driving factors from 2004 through 2012, in which years assimilated meteorological data are available to drive model simulations. Figure 9 shows that the measured annual ozone trends over 2004–2012 (Fig. 9d-f) are stronger than but spatially consistent with the trends over 1990–2014. For the US annual mean, the growth rates over 2004–2012 are 0.43 ppb/yr for the nighttime ozone and 0.14 ppb/yr for the daytime ozone, about twice the growth rates over 1990–2014. The diurnal cycle of US annual ozone trends over 2004–2012 is similar to the cycle for the 1990–2014 trends (red solid versus black solid line in Fig. 3), although with a stronger diurnal range (maximum – minimum).

We note here that the stronger trend in annual ozone over 2004–2012 is partly due to the choice of beginning and end years (Bacer et al., 2016). For example, the growth rates over 2005–2011 are 0.31 ppb/yr for the nighttime mean, 0.13 ppb/yr for the daytime mean, and 0.23 ppb/yr for the daily mean ozone (Table 2), consistent but smaller than the trends over 2004–2012. As an extreme case, the growth rates between 2002 (with a local ozone maximum) and 2014 (with a local minimum) are only 0.13 ppb/yr (P-value < 0.05) for nighttime ozone, 0.05 ppb/yr for daytime ozone, and 0.08 ppb/yr for daily mean ozone. For seasonal ozone, the trend differences between 2004–2012 and 1990–2014 are generally similar to the differences for annual ozone (Table 2). A possible reason for the stronger ozone trend in the recent decade is that anthropogenic emissions of $NO_x$ decline much more rapidly (Fig. 4b) over 2004–2012 (at a rate of 4.1%/yr relative to 2004) than over 1990–2014 (2.1%/yr relative to 1990). Also, a heat wave swept much of the US in 2012, partly contributing to the high value in that year.

### 5.1 Evaluation of modeled ozone in the "Control" simulation

Figure 10 compare the spatial distributions of modeled (the "Control" simulation) and observed 2004–2012 average daily, daytime and nighttime mean ozone over the US. The "Control" simulation overestimates the observed ozone, especially over the eastern US, a common problem in chemical transport models (Fiore et al., 2009;Lin et al., 2008;Stevenson et al., 2006;Yan et al., 2016;Young et al., 2013). Model biases are about 8.8 ppb, 6.3 ppb and 10.4 ppb for the daily, daytime and nighttime mean ozone, respectively, averaged over the US.

The solid yellow line in Fig. 11 shows that the "Control" simulation captures the diurnal variation of the observed ozone trends (red line), although with a slight systematic underestimate. The model produces significant growth in the nighttime mean ozone (0.31 ppb/yr), modest growth in

the daily mean ozone (0.22 ppb/yr), and statistically insignificant growth in the daytime ozone (0.14 ppb/yr), weaker than but consistent with the observed trends (0.19–0.43 ppb/yr). Table 5 further shows that the "Control" simulation reproduces the observed interannual and seasonal variability of ozone. The model-observation correlations (0.82–0.91, P-value < 0.01) are statistically significant for the US annual/seasonal average daily, daytime and nighttime mean ozone, no matter whether the ozone data are de-trended. The last column of Fig. 10 also shows that the "Control" simulation captures the observed interannual variability of annual ozone in most model grid cells, with statistically significant correlation coefficients.

A sensitivity simulation was conducted to test the effect of anthropogenic NMVOC emission changes not included in the "Control" simulation. In the sensitivity simulation, we scaled the NMVOC emissions to the years of 2004 and 2012 based on the EDGARv4.3.2 database which provides the emission time series (1970–2012). Emissions of NMVOC were scaled according to emissions over five regions (China, rest of Asia, the US, Europe and rest of world). Other emissions are the same with the "Control" simulation.

In this sensitivity simulation, the modeled change in annual mean ozone from 2004 to 2012 is 1.7 ppb (equivalent to 0.21 ppb/yr) averaged over the US, with local ozone changes ranging from -1.9 to 8.1 ppb across the selected 124 grid cells. This magnitude of ozone change is consistent with the "Control" simulation results with the modeled ozone trend at 0.22 ppb/yr during 2004–2012 (Table 5). For the urban sites, the mean ozone change in the sensitivity simulation (2.1 ppb, or 0.26 ppb/yr) is also close to the "Control" simulation (0.28 ppb/yr). These results suggest that changes in anthropogenic NMVOC emissions have not led to a systemic ozone trend across the US on top of the effect of NOx emission changes, consistent with the results in Simon et al. (2015).

## 5.2 Effects of anthropogenic emissions versus climate variability revealed by perturbation simulations

The second simulation (named **fixed emis** in Fig.11) tests the sole sensitivity of ozone to interannual climate variability, by fixing global anthropogenic emissions at the 2004 levels. As such, both the decline in US emissions and the growth in Asian emissions are excluded. Table 5 shows that with fixed emissions, the modeled annual daytime and daily mean ozone are still highly correlated to the observed counterparts (R = 0.61–0.83, P-value < 0.01). By comparison, the model-observation correlation becomes statistically insignificant for the annual nighttime ozone. The modeled US annual ozone trends are not statistically significant (0.05–0.07 ppb/yr) for daily mean, daytime and nighttime ozone. The short dashed yellow line in Fig. 11 also shows statistically insignificant ozone trends at individual hours when anthropogenic emissions are fixed.

For the second simulation, results for seasonal ozone are in line with those for annual ozone (Table 5). Among the seasons, the model nighttime ozone is best correlated with the observations

in summer, whereas the correlation coefficients (0.44–0.48) are much still lower than those for daytime (0.77–0.83) and daily mean (0.72–0.79) ozone. The summertime daytime ozone growth rate increases from -0.05 ppb/yr in the "Control" case to 0.03 ppb/yr, with a sign of change opposite to other seasons. This reflects the importance of controlling anthropogenic emissions to reducing summertime daytime ozone (Bloomer et al., 2009; Cooper et al., 2012; Simon et al., 2015; Sather and Cavender, 2016; Lin et al., 2017). [As discussed in Sect. 3, the summertime daytime decline here is weaker than previous results for peak ozone because we included the morning and late afternoon hours that exhibit ozone growth.] For other seasons, the ozone growth rates decrease drastically from the "Control" to the second case. This reflects "penalty" of reducing $NO_x$ (Jhun et al., 2015), which is consistent with previous findings that the 5th percentile of peak ozone (normally occurring in cold seasons) over the eastern US has increased due to weakened $NO_x$ titration (Gao et al., 2013; Clifton et al., 2014., Simon et al., 2015; Lin et al., 2017).

The third sensitivity simulation fixes the US anthropogenic emissions at the 2004 levels while allowing emissions in other regions and meteorology to vary interannually (Table 5). The resulting ozone growth rates and model-observation correlations resemble the second case, suggesting that reductions in the US $NO_x$ emissions and ozone titration are the dominant driver of modeled all-season nighttime ozone growth and summertime daytime ozone reduction over 2004–2012. For the US annual ozone, the effects of decreasing US emissions are approximately 0.15 ppb/yr for daily mean ozone, 0.08 ppb/yr for daytime mean ozone and 0.22 ppb/yr for nighttime mean ozone, based on the difference between the third and the "Control" simulation. The annual ozone growth rates are slightly higher (by 0.01–0.02 ppb/yr) than the second case because of rising Asian emissions simulated in the third case but not in the second case. Seasonally, the increase from the second to the third case is greatest in spring (0.11–0.14 ppb/yr versus 0.06–0.07 ppb/yr, Table 5). The contribution of rising Asian emissions to the springtime US ozone growth, especially over the western US, were also found by previous studies (Fiore et al., 2009; Huang et al., 2013; Lin et al., 2015; Verstraeten et al., 2014; Lin et al., 2017).

The fourth simulation (named **fixed met** in Fig.11) fixes meteorological data in 2004 but allows global anthropogenic emissions to vary interannually. The resulting trends of annual ozone are close to the trends in the "Control" simulation across the individual hours (long dashed yellow versus solid yellow line in Fig. 11). Table 5 shows that the trend in annual nighttime mean ozone is still notable, at 0.26 ppb/yr (P-value < 0.01) compared to 0.31 ppb/yr in the "Control" simulation. This confirms that the nighttime growth is driven by reduced $NO_x$ emissions and weakened ozone titration. The model-observation correlation for the annual nighttime ozone is 0.71 (P-value < 0.01) and 0.31 (P-value = 0.43) before and after de-trending, respectively. The correlations for the annual daytime ozone are much weaker than the second simulation no matter whether the ozone data are de-trended (0.39–0.41 versus 0.76–0.81), suggesting the dominant effect of interannual climate variability on the ozone in this part of the day. For season ozone, changes from the "Control" to the fourth case are generally similar to those for annual ozone (Table 5).

For each hour of the day, the two shaded areas in Fig. 11 broadly separate the contribution of anthropogenic emissions (dark grey shade) to the 2004–2012 annual ozone changes from the contribution of interannual climate variability (light grey shade). The contributions are calculated as $C_{anth} = R_{anth}^2 / (R_{anth}^2 + R_{clim}^2)$ and $C_{clim} = 1 - C_{anth}$, where $R_{anth}$ is the correlation between observed and modeled annual mean ozone (at a particular hour) with fixed model meteorology and $R_{clim}$ the observation-model correlation with fixed global anthropogenic emissions. Figure 11 shows that the emission contribution dominates in the nighttime hours (relatively constant at about 70%), with a reduction in the morning hours, an increase in the late afternoon hours, and a minimum value (at 10%) around 15:00.

Overall, the modeling results indicate that for the interannual variation of US mean ozone over 2004–2012, climate variability and anthropogenic emissions are the main drivers of the historical daytime ozone variability and nighttime ozone trend, respectively. The rising Asian emissions contribute to the US annual mean ozone trends (0.01–0.02 ppb/yr for daily mean, daytime and nighttime ozone) much less than the contributions from the US anthropogenic emission changes (0.08–0.22 ppb/yr) and climate variability (0.05–0.07 ppb/yr).

**6. Concluding Remark**

This work shows that reductions in the US anthropogenic emissions have effectively lowered the summertime daytime ozone from 1990 to 2014, consistent with previous studies on afternoon or MDA8 ozone. On an annual mean basis, the daytime ozone have continued to increase. Furthermore, the great sensitivity of the annual average daytime ozone to interannual climate variability increases the difficulty in projecting future ozone air quality (Jacob and Winner, 2009;Rieder et al., 2015). The daily mean and particularly the nighttime ozone have experienced substantial growth, due to weakened titration by $NO_x$. This likely implies potential growth in health risk from long-term exposure of enhanced low- and medium-level ozone (Jerrett et al., 2009;Bell et al., 2006;Peng et al., 2013). As the extent of outdoor activities differs notably at different times of day, the overall effect of ozone trends at individual hours on public health warrants further research. Nonetheless, pollution mitigation strategies might consider to address ozone changes at different times of the day and not only during peak hours.

**Acknowledgements**

We thank Owen Cooper, Haidong Kan and Jianjun Yin for discussions. This research is supported by the 973 project (2014CB441303) and the National Natural Science Foundation of China (41775115 and 41422502). We acknowledge the free use of ozone data from AQS (http://aqsdr1.epa.gov/aqsweb/aqstmp/airdata/download_files.html), the AMO index from NOAA/ESRL (http://www.esrl.noaa.gov/psd/data/timeseries/AMO/), and the ONI index from the NOAA Climate Prediction Center (http://www.cpc.noaa.gov/products/analysis_monitoring/ensostuff/ensoyears.shtml).

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

| Data selection criteria | Ozone trend (ppb/yr) and correlation | | | | | | | | |
|---|---|---|---|---|---|---|---|---|---|
| | Daily | | | Daytime | | | Nighttime | | |
| | Trend (ppb/yr) | Corr.[7] | De-trended Corr.[8] | Trend (ppb/yr) | Corr.[7] | De-trended Corr.[8] | Trend (ppb/yr) | Corr.[7] | De-trended Corr.[8] |
| Default [2] | $0.16^{**}$ | | | $0.09^{*}$ | | | $0.21^{**}$ | | |
| Default_strict [3] | $0.15^{**}$ | $0.94^{**}$ | $0.94^{**}$ | $0.08^{*}$ | $0.93^{**}$ | $0.94^{**}$ | $0.21^{**}$ | $0.95^{**}$ | $0.93^{**}$ |
| Data-continuity [4] | $0.17^{**}$ | $0.88^{**}$ | $0.87^{**}$ | $0.09^{*}$ | $0.87^{**}$ | $0.87^{**}$ | $0.24^{**}$ | $0.88^{**}$ | $0.86^{**}$ |
| Data-coverage [5] | $0.14^{**}$ | $0.86^{**}$ | $0.85^{**}$ | $0.06^{*}$ | $0.83^{**}$ | $0.85^{**}$ | $0.22^{**}$ | $0.87^{**}$ | $0.84^{**}$ |
| Cooper et al. [6] | $0.15^{**}$ | $0.88^{**}$ | $0.87^{**}$ | $0.06^{*}$ | $0.85^{**}$ | $0.87^{**}$ | $0.23^{**}$ | $0.89^{**}$ | $0.87^{**}$ |

1.   ** P-value < 0.01, * P-value < 0.05 under an *F*-test for trends and a one-sided *T*-test for
correlation.
2.   based on data in the 124 grid cells, our default choice.
3.   similar to a, but based on data in the 100 grid cells that include at least three sites.
4.   based on data in the 94 grid cells covering the sites with valid data in at least three years for
every five years.
5.   based on 70 sites with more than 75% of hourly data available in all years.
6.   based on 82 sites passing criteria similar to those adopted by Cooper et al.(Cooper et al.,
2012).
7.   Correlation between the US annual ozone time series in a sensitivity case and the time series
in the default case.
8.   Similar to 7 but for de-trended ozone.

Table 2. Observed trends for seasonal and annual ozone over the eastern (25 °–50 °N, 65 °–100 °W)
and western (25 °–50 °N, 100 °–125 °W) United States during various time periods [1].

|  | MAM | JJA | SON | DJF | Annual |
|---|---|---|---|---|---|
| Daily mean ozone trend (ppb/yr) | | | | | |
| 1990–2014 US | 0.21$^{**}$ | 0.02 | 0.14$^{**}$ | 0.13$^{**}$ | 0.16$^{**}$ |
| 1990–2014 Eastern US | 0.19$^{**}$ | -0.03 | 0.12$^{**}$ | 0.12$^{**}$ | 0.12$^{**}$ |
| 1990–2014 Western US | 0.25$^{**}$ | 0.12$^{**}$ | 0.16$^{**}$ | 0.16$^{**}$ | 0.20$^{**}$ |
| 2004–2012 US | 0.34$^{**}$ | 0.16$^{**}$ | 0.29$^{**}$ | 0.27$^{**}$ | 0.30$^{**}$ |
| 2005–2011 US | 0.27$^{**}$ | 0.09$^{*}$ | 0.20$^{**}$ | 0.21$^{**}$ | 0.23$^{**}$ |
| 2002–2014 US | 0.12$^{**}$ | 0.04 | 0.07$^{*}$ | 0.06 | 0.08$^{*}$ |
| Daytime mean ozone trend (ppb/yr) | | | | | |
| 1990–2014 US | 0.17$^{**}$ | -0.08$^{*}$ | 0.09$^{*}$ | 0.12$^{**}$ | 0.09$^{*}$ |
| 1990–2014 Eastern US | 0.15$^{**}$ | -0.11$^{*}$ | 0.07$^{*}$ | 0.11$^{*}$ | 0.06 |
| 1990–2014 Western US | 0.21$^{**}$ | 0.03 | 0.10$^{*}$ | 0.15$^{**}$ | 0.13$^{**}$ |
| 2004–2012 US | 0.26$^{**}$ | -0.03 | 0.20$^{**}$ | 0.24$^{**}$ | 0.19$^{**}$ |
| 2005–2011 US | 0.21$^{**}$ | -0.04 | 0.14$^{**}$ | 0.16$^{**}$ | 0.13$^{**}$ |
| 2002–2014 US | 0.09$^{*}$ | 0.03 | 0.05 | 0.06 | 0.05 |
| Nighttime mean ozone trend (ppb/yr) | | | | | |
| 1990–2014 US | 0.26$^{**}$ | 0.13$^{**}$ | 0.19$^{**}$ | 0.14$^{**}$ | 0.21$^{**}$ |
| 1990–2014 Eastern US | 0.24$^{**}$ | 0.05 | 0.16$^{**}$ | 0.13$^{**}$ | 0.18$^{**}$ |
| 1990–2014 Western US | 0.30$^{**}$ | 0.20$^{**}$ | 0.23$^{**}$ | 0.17$^{**}$ | 0.25$^{**}$ |
| 2004–2012 US | 0.46$^{**}$ | 0.35$^{**}$ | 0.40$^{**}$ | 0.30$^{**}$ | 0.43$^{**}$ |
| 2005–2011 US | 0.35$^{**}$ | 0.24$^{**}$ | 0.28$^{**}$ | 0.25$^{**}$ | 0.31$^{**}$ |
| 2002–2014 US | 0.16$^{**}$ | 0.06 | 0.11$^{*}$ | 0.07$^{*}$ | 0.13$^{**}$ |

1.  ** P-value < 0.01. * P-value < 0.05 under an *F*-test.

1    Table 3. Observed trends for the $5^{th}$, $50^{th}$ and $95^{th}$ percentiles of ozone over the rural, suburban
2    and urban areas during 1990–2014 [1].

|  | Rural | Suburban | Urban |
|---|---|---|---|
| Daily mean ozone trend (ppb/yr) | | | |
| $5^{th}$ | $0.25^{**}$ | $0.25^{**}$ | $0.24^{**}$ |
| $50^{th}$ | $0.11^{*}$ | $0.15^{**}$ | $0.21^{**}$ |
| $95^{th}$ | 0.06 | 0.04 | $0.08^{*}$ |
| Daytime mean ozone trend (ppb/yr) | | | |
| $5^{th}$ | $0.29^{**}$ | $0.29^{**}$ | $0.28^{**}$ |
| $50^{th}$ | 0.02 | $0.10^{*}$ | $0.19^{**}$ |
| $95^{th}$ | $-0.13^{**}$ | $-0.21^{**}$ | $-0.15^{**}$ |
| Nighttime mean ozone trend (ppb/yr) | | | |
| $5^{th}$ | $0.21^{**}$ | $0.21^{**}$ | $0.19^{**}$ |
| $50^{th}$ | $0.19^{**}$ | $0.20^{**}$ | $0.22^{**}$ |
| $95^{th}$ | $0.22^{**}$ | $0.24^{**}$ | $0.27^{**}$ |

3    1.  ** P-value < 0.01. * P-value < 0.05 under an *F*-test.

1    Table 4. Correlations (R) between de-trended time series of AMONI index and de-trended daily
2    mean ozone, daytime mean ozone and nighttime mean ozone in different seasons and regions [1].

|  |  | MAM | JJA | SON | DJF | Annual |
|---|---|---|---|---|---|---|
| US | Daily | 0.25 | 0.58$^*$ | 0.67$^{**}$ | 0.22 | 0.62$^{**}$ |
|  | Daytime | 0.19 | 0.67$^{**}$ | 0.70$^{**}$ | 0.23 | 0.71$^{**}$ |
|  | Nighttime | 0.29 | 0.51$^*$ | 0.59$^*$ | 0.21 | 0.33 |
| Eastern US | Daily | 0.11 | 0.66$^{**}$ | 0.68$^{**}$ | 0.06 | 0.68$^{**}$ |
|  | Daytime | 0.11 | 0.72$^{**}$ | 0.74$^{**}$ | 0.18 | 0.73$^{**}$ |
|  | Nighttime | 0.07 | 0.55$^*$ | 0.60$^{**}$ | 0.09 | 0.38 |
| Western US | Daily | 0.48$^*$ | 0.27 | 0.22 | 0.44$^*$ | 0.45$^*$ |
|  | Daytime | 0.38 | 0.25 | 0.19 | 0.37 | 0.34 |
|  | Nighttime | 0.50$^*$ | 0.27 | 0.24 | 0.46$^*$ | 0.46$^*$ |

3    1.** for P-value < 0.01 and * for P-value < 0.05 under a one-side $T$-test

Table 5. Interannual variability of observed and modeled ozone during 2004–2012 and their correlation[1].

| | Observation | Model | | | |
|---|---|---|---|---|---|
| | | Control | Fixed global anthropogenic emissions | Fixed US anthropogenic emissions | Fixed meteorology |
| **Annual daily mean ozone** | | | | | |
| Trend (ppb/yr) | 0.30** | 0.22** | 0.06 | 0.07 | 0.18** |
| Correlation | | 0.86** | 0.64** | 0.65** | 0.52* |
| De-trended correlation[2] | | 0.89** | 0.61** | 0.63** | 0.31 |
| **Annual daytime mean ozone** | | | | | |
| Trend (ppb/yr) | 0.19** | 0.14 | 0.05 | 0.06 | 0.10 |
| Correlation | | 0.88** | 0.76** | 0.78** | 0.41* |
| De-trended correlation[2] | | 0.90** | 0.81** | 0.82** | 0.39 |
| **Annual nighttime mean ozone** | | | | | |
| Trend (ppb/yr) | 0.43** | 0.31** | 0.07 | 0.09 | 0.26** |
| Correlation | | 0.86** | 0.39 | 0.37 | 0.71** |
| De-trended correlation[2] | | 0.88** | 0.24 | 0.28 | 0.31 |
| **Spring daily mean ozone** | | | | | |
| Trend (ppb/yr) | 0.34** | 0.25** | 0.07 | 0.12 | 0.15** |
| Correlation | | 0.84** | 0.62** | 0.67** | 0.61** |
| De-trended correlation[2] | | 0.87** | 0.65** | 0.69** | 0.42 |
| **Spring daytime mean ozone** | | | | | |
| Trend (ppb/yr) | 0.26** | 0.19** | 0.06 | 0.11 | 0.09 |
| Correlation | | 0.89** | 0.73** | 0.76** | 0.52* |
| De-trended correlation[2] | | 0.90** | 0.78** | 0.77** | 0.38 |
| **Spring nighttime mean ozone** | | | | | |
| Trend (ppb/yr) | 0.46** | 0.32** | 0.07 | 0.14 | 0.20** |
| Correlation | | 0.83** | 0.41* | 0.49* | 0.69** |
| De-trended correlation[2] | | 0.85** | 0.33 | 0.36 | 0.34 |
| **Summer daily mean ozone** | | | | | |
| Trend (ppb/yr) | 0.16* | 0.09 | 0.04 | 0.03 | 0.06 |
| Correlation | | 0.88** | 0.72** | 0.77** | 0.50* |
| De-trended correlation[2] | | 0.89** | 0.79** | 0.79** | 0.28 |
| **Summer daytime mean ozone** | | | | | |
| Trend (ppb/yr) | -0.03 | -0.05 | 0.03 | 0.02 | -0.07 |
| Correlation | | 0.89** | 0.77** | 0.77** | 0.48* |
| De-trended correlation[2] | | 0.91** | 0.83** | 0.81** | 0.35 |
| **Summer nighttime mean ozone** | | | | | |
| Trend (ppb/yr) | 0.35** | 0.24** | 0.05 | 0.05 | 0.18* |
| Correlation | | 0.86** | 0.48* | 0.46* | 0.58* |
| De-trended correlation[2] | | 0.88** | 0.44* | 0.48* | 0.40 |
| **Fall daily mean ozone** | | | | | |
| Trend (ppb/yr) | 0.29** | 0.22** | 0.06 | 0.06 | 0.19** |
| Correlation | | 0.85** | 0.67** | 0.66** | 0.56* |
| De-trended correlation[2] | | 0.87** | 0.65** | 0.67** | 0.32 |
| **Fall daytime mean ozone** | | | | | |
| Trend (ppb/yr) | 0.20** | 0.13 | 0.05 | 0.06 | 0.11 |
| Correlation | | 0.86** | 0.69** | 0.68** | 0.49* |
| De-trended correlation[2] | | 0.87** | 0.72** | 0.71** | 0.34 |
| **Fall nighttime mean ozone** | | | | | |
| Trend (ppb/yr) | 0.40** | 0.29** | 0.07 | 0.07 | 0.26** |
| Correlation | | 0.84** | 0.36 | 0.37 | 0.67** |
| De-trended correlation[2] | | 0.87** | 0.29 | 0.33 | 0.36 |
| **Winter daily mean ozone** | | | | | |
| Trend (ppb/yr) | 0.27** | 0.21** | 0.05 | 0.07 | 0.19** |
| Correlation | | 0.83** | 0.61** | 0.63** | 0.61** |
| De-trended correlation[2] | | 0.84** | 0.63** | 0.64** | 0.33 |
| **Winter daytime mean ozone** | | | | | |
| Trend (ppb/yr) | 0.24** | 0.17* | 0.04 | 0.05 | 0.12 |
| Correlation | | 0.85** | 0.68** | 0.67** | 0.54* |
| De-trended correlation[2] | | 0.86** | 0.72** | 0.73** | 0.32 |
| **Winter nighttime mean ozone** | | | | | |

| | | | | | |
|---|---|---|---|---|---|
| Trend (ppb/yr) | 0.30** | 0.24** | 0.05 | 0.08 | 0.25** |
| Correlation | | 0.82** | 0.31 | 0.35 | 0.72** |
| De-trended correlation[2] | | 0.84** | 0.23 | 0.26 | 0.29 |

1. ** P-value < 0.01, * P-value < 0.05 under an *F*-test for trends and a one-sided *T*-test for correlation.
2. Model and observation data are de-trended prior to correlation calculations.

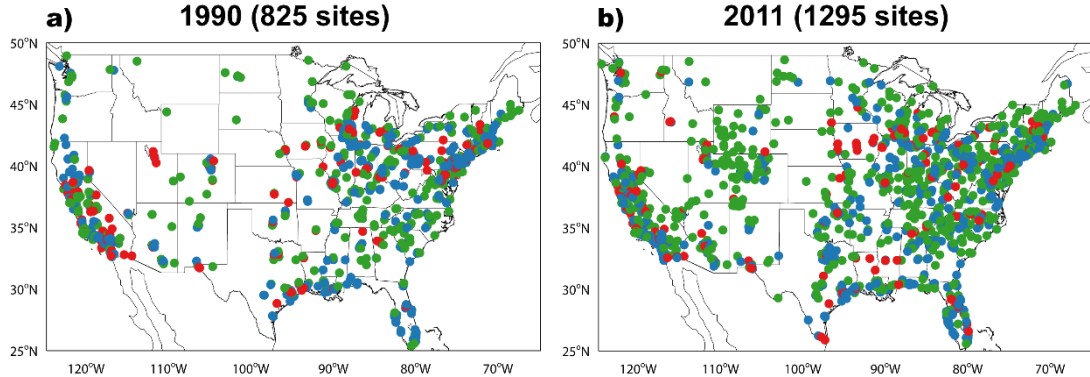

Figure 1. AQS ozone site distributions in 1990 (with the fewest sites) and 2011 (with the most sites). Rural, suburban and urban sites are shown in green, blue and red, respectively.

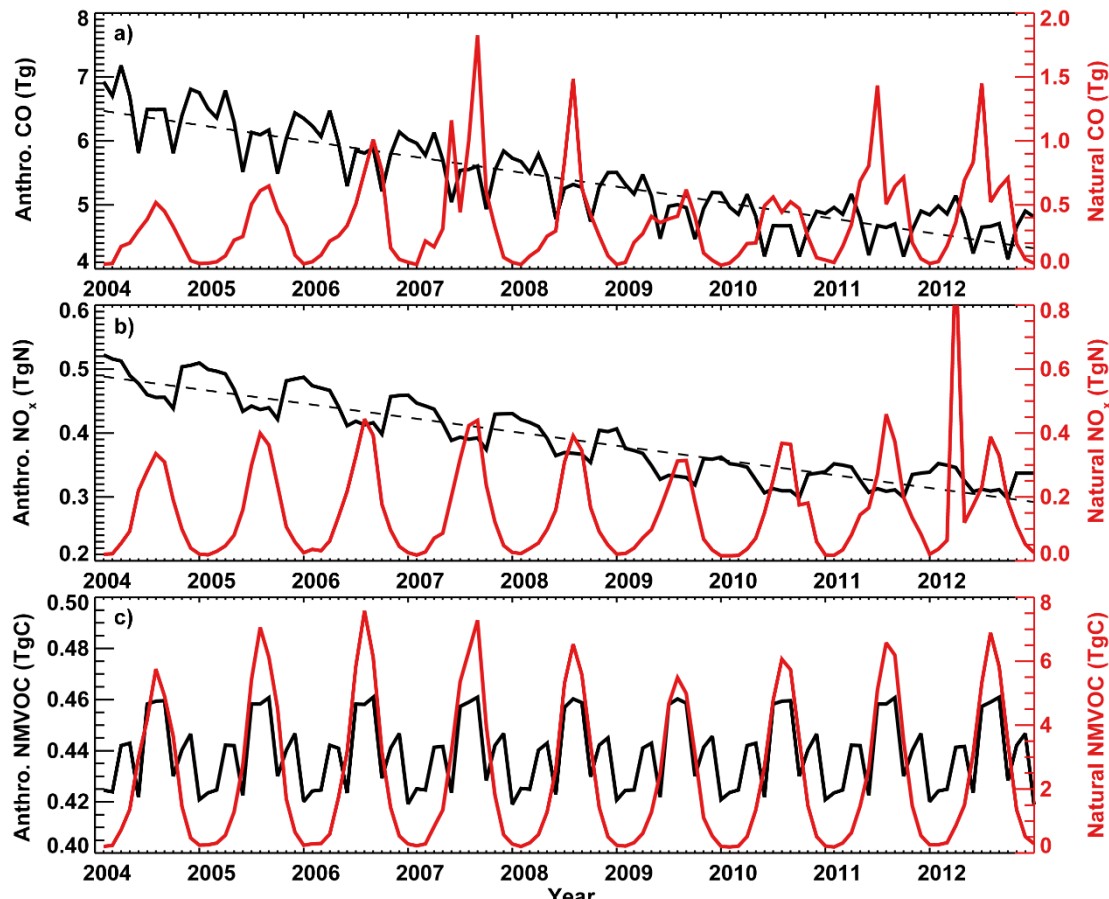

Figure 2. Monthly anthropogenic (fossil + biofuel, black lines) and natural (red lines) emissions
of ozone precursors over the US used in GEOS-Chem. Natural $NO_x$ emissions include biomass
burning, lightning, and soil (including fertilizer) sources. Natural CO emissions are from biomass
burning and oxidization of monoterpenes. Natural NMVOC emissions are from biomass burning
and biogenic sources.

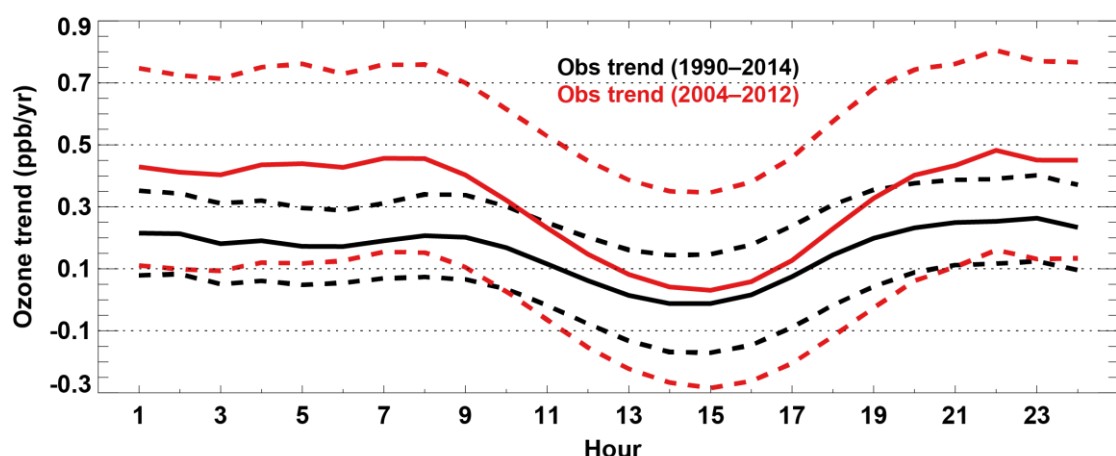

3 Figure 3. Trends in the observed surface ozone over the US, calculated based on data in the
4 selected 124 grid cells. The black line shows the 1990–2014 trend in the US annual mean ozone
5 for each hour of the day (local standard time), the red line depicts the observed trend over 2004–
6 2012, and the dashed lines indicate their deviations.

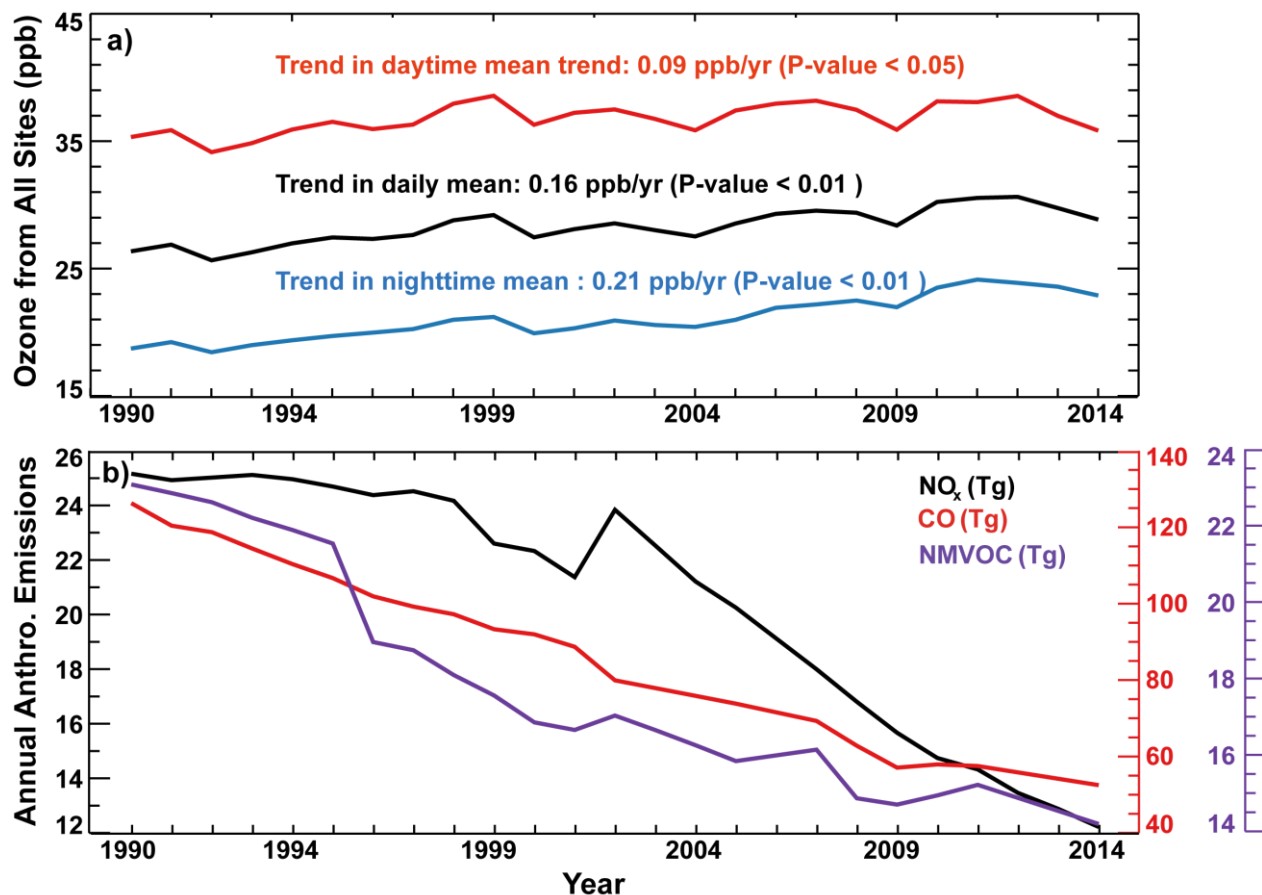

Figure 4. Long-term trends in the observed surface ozone and anthropogenic precursor emissions over the US during 1990–2014. (a) The ozone trend averaged over the 124 gridd cells. (b) The US annual anthropogenic (fossil + biofuel) emissions of $NO_x$ (black line), CO (red line), and NMVOC (purple line) from the National Emissions Inventory (http://www.epa.gov/air-emissions-inventories/air-pollutant-emissions-trends-data).

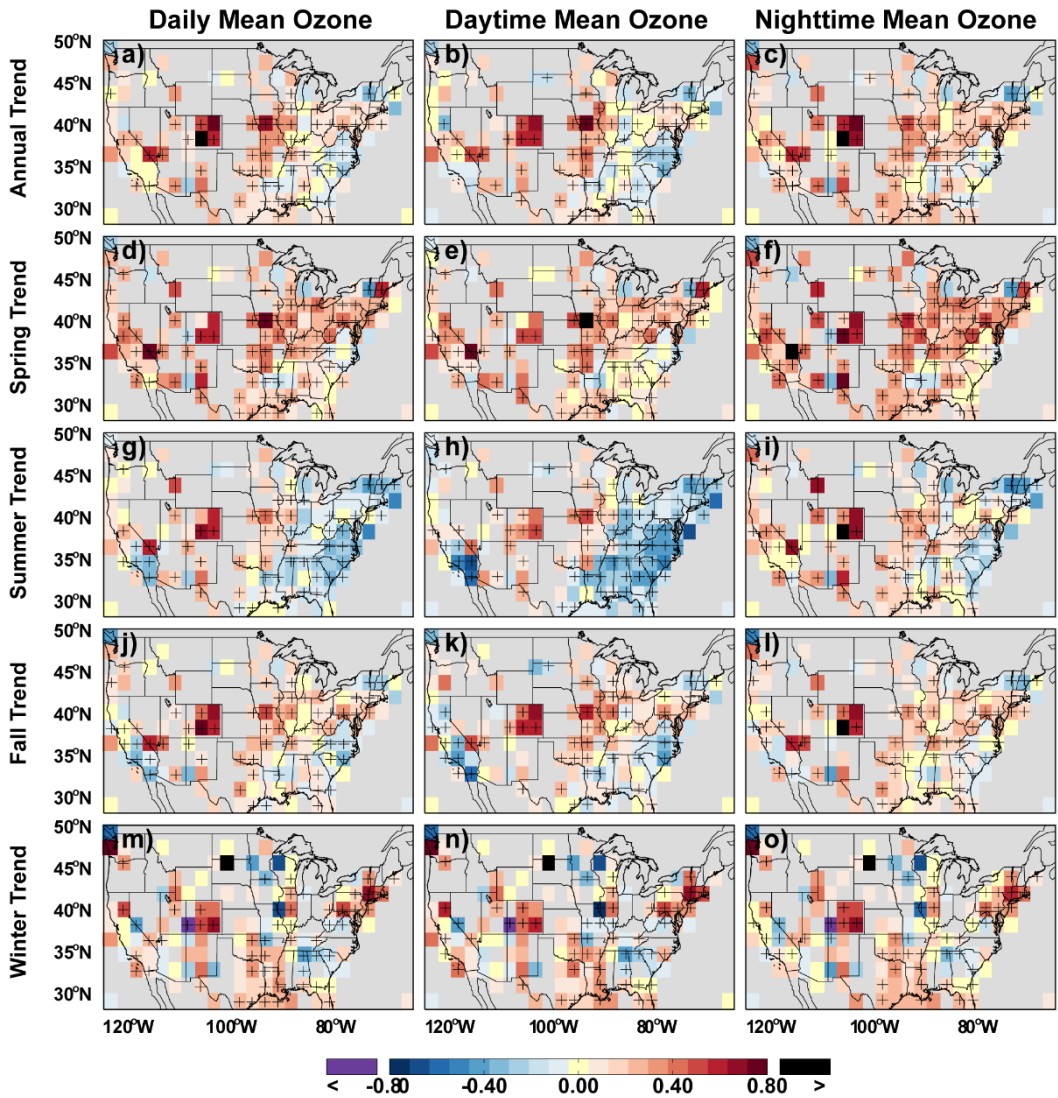

Figure 5. Trends over 1990–2014 in annual and seasonal daily mean, daytime mean, and
nighttime mean ozone observations in the selected 124 grid cells. Trends are statistically
significant (P-value $< 0.05$ under a $F$-test) in grid cells overlaid with '+'. The grid cells in grey
have no data.

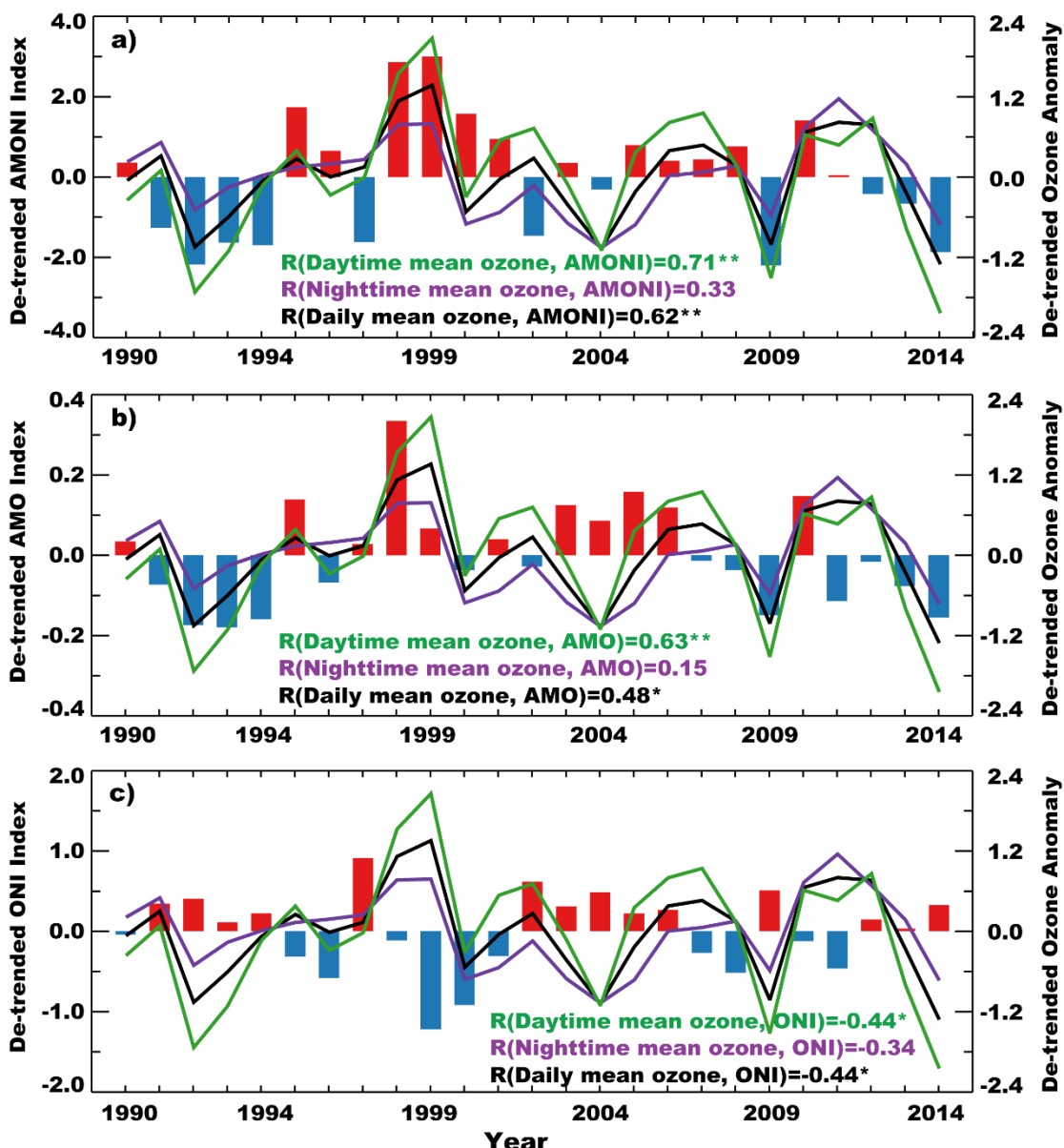

Figure 6. De-trended anomaly time series of climate indices and observed US mean ozone. The anomalies of annual mean daily mean ozone (black lines), daytime mean ozone (green lines) and nighttime mean ozone (purple lines) are plotted with the annual AMONI index (**a**), with the annual AMO index (**b**), and with the annual ONI index (**c**). Red and blue bars indicate positive and negative anomalies of climate indices, respectively. Also shown are correlations between ozone and individual indices ('**\*\***' for P-value < 0.01 and '**\***' for P-value < 0.05 under a one-side *T*-test).

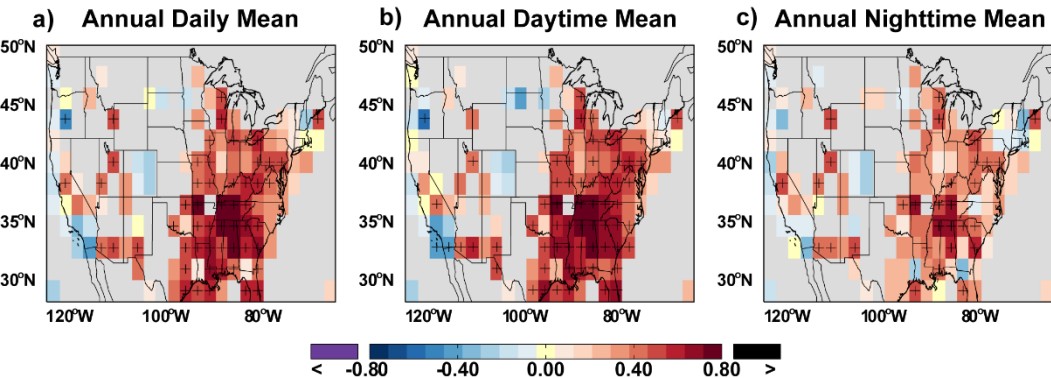

Figure 7. Correlation in interannual variability over 1990–2014 between the annual AMONI index and the annual average daily mean ozone (a), daytime mean ozone (b), and nighttime mean ozone (c). All data are de-trended prior to correlation calculations. Correlations are statistically significant (P-value < 0.05 under a one-sided *T*-test) in grid cells overlaid with '+'. The grid cells in grey have no AQS data.

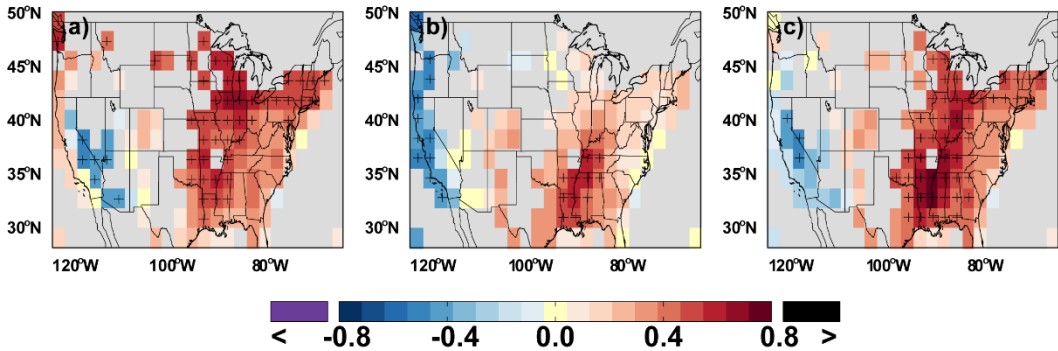

Figure 8. Correlation for interannual variability over 1990–2010 between the annual AMO (a),
ONI (b), and AMONI (c) anomalies and the annual average daily mean 2-meter air temperature
from MERRA. Correlations are statistically significant (P-value < 0.05 under a one-sided *T*-test)
in grid cells overlaid with '+'. MERRA temperature data are available through 2010, and are
sampled based on valid daily mean ozone data. MERRA data are used here (in place of GEOS-5)
to include years prior to 2004. All data are de-trended prior to correlation calculations. The grid
cells in grey have no AQS data.

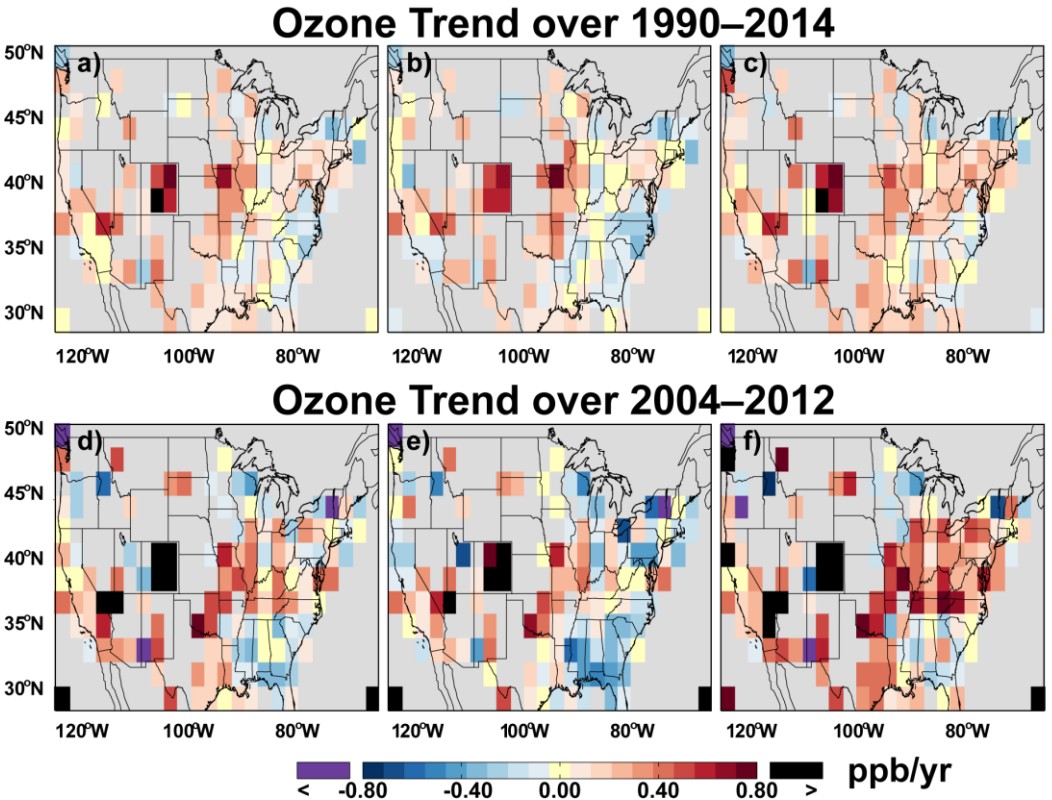

Figure 9. Trends in AQS annual ozone over different time periods. (a-c) Trends (ppb/yr) over
1990–2014 in daily mean ozone (a), daytime mean ozone (b), and nighttime mean ozone (c). (d-f)
Similar to (a-c) but for trends over 2004–2012. Panels (a-c) are the same as Fig. 5a-c. The grid
cells in grey have no data.

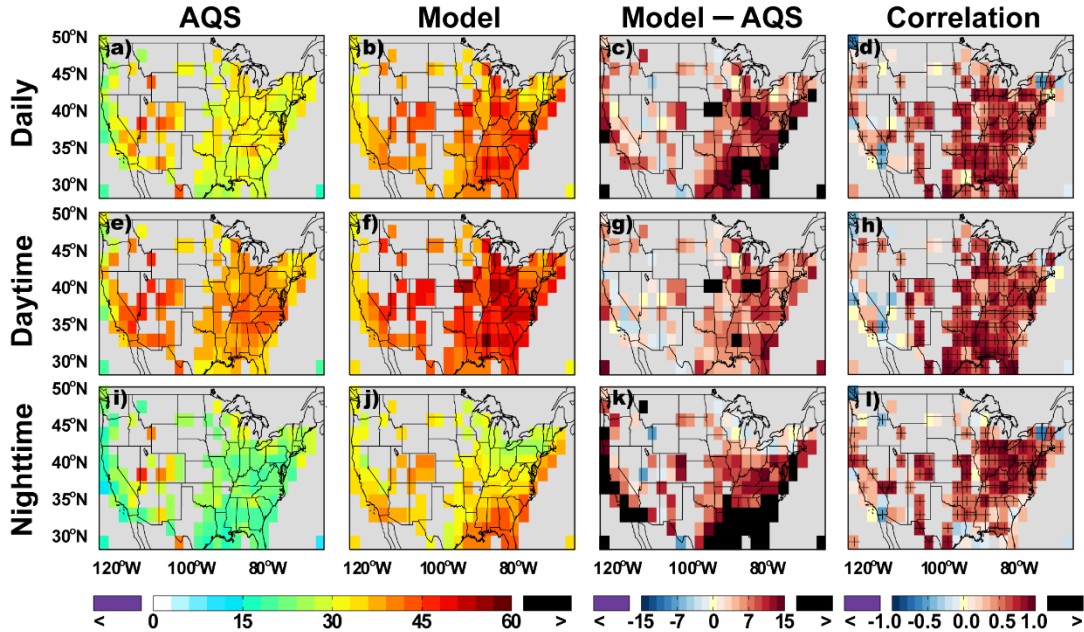

Figure 10. Observed and modeled 2004–2012 average daily, daytime and nighttime mean ozone
(ppb) over the US, as well as model biases (ppb) against and interannual correlation to the
observations. Grid cells with statistically significant correlation coefficients are highlighted by
'+'. The grid cells in grey have no AQS data.

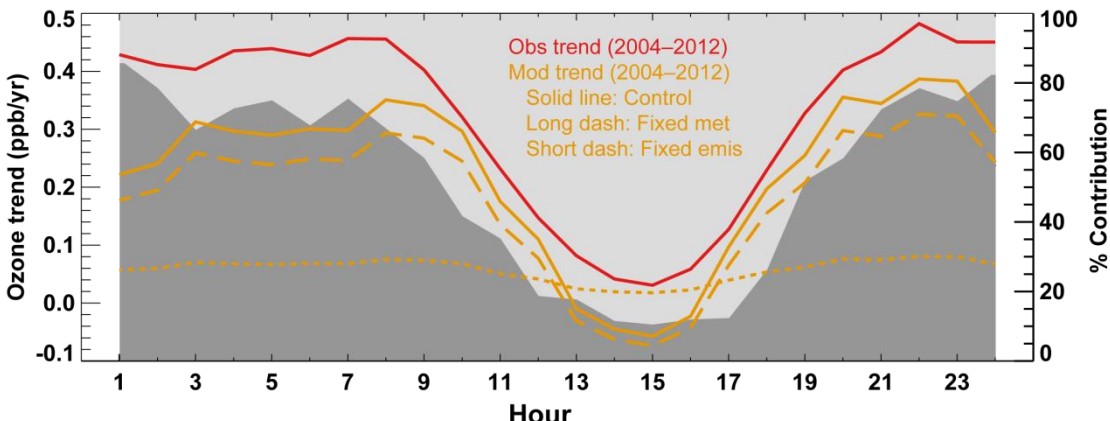

Figure 11. Trends in the modeled surface ozone over the US at different times of day. The yellow
lines depict the trends in three model simulations (control, fixed meteorology, and fixed global
anthropogenic emissions). The two shaded areas indicate the individual contributions of
interannual climate variability (light grey shade) and global anthropogenic emissions (dark grey
shade) to the simulated 2004–2012 ozone changes.

