# Peer review of "Ozone trends over the United States at different times of day"

_Atmospheric Chemistry and Physics, 2017_

## Referee Comment (RC2)

The authors have conducted a very interesting study to investigate the ozone trends at different times of a day. This work includes many important details and could have important implications for future ozone air quality control. So I recommend this paper could be published after they have addressed some issues as listed below.

Major comments.

When trying to quantify the influence of meteorology on U.S. ozone trends during 1990-2014, the authors focused on two climate patterns—AMO and ONI. A new index named AMONI is also constructed. Even they have found strong correlations between the annual mean ozone with these climate indices, but the underlying physical mechanism is unclear to the readers. It may be too reckless to say US ozone is dependent on AMONI. Here are my concerns.

1. According to a new study (Shen et al., 2017), the U.S. summertime ozone is associated with a tripole SST pattern in the Atlantic and a dipole pattern in the northeastern Pacific. This raises the question that why the authors choose to use AMO and ONI. Is this the best choice?

2. The mechanism related to ENSO as proposed by Xu et al. [2017] only applies to the spring or fall season (e.g. the subtropical jet wind only explains the springtime ozone decreases in a small region in the southwest US). Since annual mean ozone is used in this study, the authors can't simply use the mechanism in Xu et al. [2017] to support their conclusion here.

3. The influence of AMO on U.S. weather varies by season and region. See Sutton et al. [2007] for more details. Also, AMO not only influence the temperature but also the subsidence, precipitation, drought and surface wind. Throughout this paper, the authors only mention the impact on temperature, which is not enough.

4. Does annual mean ozone strongly depend on temperature? It seems the authors mainly mention the influence of temperature changes when they explain the relationship of AMONI and annual mean ozone.

Minor comments.

P1 L32. Should mention that his mechanism only applies to the surface ozone.

P1 L35. We usually use "MDA8" instead of "DM8A".

P2 L1-7. The summary here is too general. Try to give more details or you can just cite two review papers (Jacob and Winner, 2009; Fiore et al., 2015).

P2 L19-24. Move this paragraph to the last paragraph of the introduction part.

P2 L25-34. These studies choose to focus on one region or one season due to potential occurrence of ozone episodes. So why should we care about the low ozone part, e.g. the ozone in the wintertime?

P3 L8-9. What is the meaning of "21.3-28.5% of data are missing"? It is unclear to the readers.

P4 L13-15. Please make it clear why you choose these two climate indices?

P4 L19-22. The summary of the influence of AMO on U.S. weather is too general.

P4 L16-23. What definition of AMO is used here? What SST product (ERSST V4 or HadISST?) is used to calculate the AMO index?

Does Fu 2015 mention that AMO can enhance the temperature anomaly and ozone production in US? They just speculate that AMO could change the ozone transport

between the north and south US. Also to my knowledge, the influence of AMO on US weather varies in different seasons. Please refer to Sutton et al. [2007].

P4 L24-30. What SST product is used here? Why the authors choose to use Nino 3.4 rather than other indices like Nino 1+2, Nino3 or Nino 4?

P4 L31-36. Please specify the detrending method here. Is there a reason that you want to subtract the influence of ONI from AMO?

P8 L33. Change "Relation" to "Relationship".

P8 L35-37. Why do you use linear detrending given the fact that the domestic NOx emissions are not linearly decreased?

P9 L13-16. The evidence shown here is not sufficient to support that positive AMONI leads to enhanced temperatures. This may be just a coincidence. None of the three references listed here can support the conclusion here.

P9 L17. Can the mechanism proposed in Xu et al. [2017] be able to explain the relationship? Xu's mechanism only applies to the springtime ozone decreases in a small region in the southwest US, while the authors here are discussing the annual mean ozone.

P9 L26-29. It is good to try AO, PDO and NAO. But there are also a lot of other climate indices. This doesn't mean AMONI is the best. See the metrics used in Shen et al. [2017].

P10 L16-24. The reason is that a heatwave swept across much of the US in 2012. The authors may need to mention this here.

P11 L9. Do you use the emissions in 2004 here?

P12 L19-27. I appreciate the author's efforts in quantifying the contribution of anthropogenic emissions and meteorological variability. But are the influences of these two processes independent? Why R is used here? Is it better to use $R^2$?

Reference:

Fiore, A. M., V. Naik, and E. M. Leibensperger (2015), Air quality and climate connections, J. Air Waste Manage. Assoc., 65(6), 645–685, doi:10.1080/10962247.2015.1040526.

Sutton RT, Hodson DL (2007) Climate response to basin-scale warming and cooling of the North Atlantic Ocean. J Climate, 20: 891-907.

Shen, L. and L. J. Mickley (2017a), Influence of large-scale climate patterns on summertime U.S. ozone: A seasonal predictive model for air quality management, Proc. Natl. Acad. Sci. U.S.A., 114(10), 2491–2496.

---

## Referee Comment (RC1) · Anonymous Referee #1 · 3 Sep 2017

The authors analyzed the inter-annual variability and trends of daytime, nighttime and daily mean ozone during 1990-2014 over the United States based on air quality monitoring data at about 1000 stations, and also assessed the impacts of anthropogenic emissions versus climate variability on the ozone trends during 2004-2012 by the GEOS-Chem modeling. This work combines observations and global modeling to evaluate the ozone trends and driving factors in the past two decades over US, both diurnally and spatially, and provide useful information about the ozone trends at the non-peak hours. The manuscript is clearly organized and well written, and the interpretation of the observational and modeling results is also fairly well. I recommend that this paper can be considered for publication after the following comments being addressed.

Specific comments:

1. On the "Control" simulation: the model only accounted for the inter-annual variations of emission inventory for NOx and CO, but ignored that for NMVOCs. The authors argued that the US anthropogenic NMVOC emissions are much smaller than the natural ones and are hence negligible. The effect of the reduction in anthropogenic NMVOC emissions in US on the ozone trends should be evaluated, especially in the urban areas. Furthermore, the changes of the NMVOC emissions in other regions (ca. Europe and Asia) should be also taken into account, as it may influence the subsequent modeling estimation of the Asian contribution to the US ozone trend. At least, sensitivity modeling studies should be done to check if considering the changes of NMVOC emissions in different regions could affect the conclusions of the modeling study.

2. Figure 3: the ozone growth rate during 2004-2012 is much faster than that in 1990-2014, especially during nighttime hours. The authors explain this in Section 5 partly due to the choice of beginning and end years. Is there any other reason for the stronger ozone trend in the recent decade?

3. Section 3: it would be better if the authors can compare the observed ozone trends in US with those from other regions of the world, such as Europe and Asia.

4. Section 4: the analyses revealed the weaker correlations between climate variability and the nighttime ozone anomaly (compared to the daytime and daily average), and between climate variability and ozone anomalies over western US (compared to eastern US). The authors need comment on the possible reasons for the weak relationships between climate and nighttime ozone, and western US ozone.

5. Page 12, Line 30-31: may the authors comment on the significance, quantitatively, of the Asian contribution in comparison with those from US and climate variability?

6. Tables 1-3: provide the unit, ppbv yr-1?

7. Table 4: what do the numbers in this table mean? Correlation coefficient (r or r2)?

8. Figure 4a: provide p-values for the trends.

9. Page 6, Line 32: delete one "mean".

СЗ

---

## Author Comment (AC1) · 23 Nov 2017

Referee #1

The authors analyzed the inter-annual variability and trends of daytime, nighttime and daily mean ozone during 1990-2014 over the United States based on air quality monitoring data at about 1000 stations, and also assessed the impacts of anthropogenic emissions versus climate variability on the ozone trends during 2004-2012 by the GEOS-Chem modeling. This work combines observations and global modeling to evaluate the ozone trends and driving factors in the past two decades over US, both diurnally and spatially, and provide useful information about the ozone trends at the non-peak hours. The manuscript is clearly organized and well written, and the inter-

pretation of the observational and modeling results is also fairly well. I recommend that this paper can be considered for publication after the following comments being addressed.

We thank the reviewer for comments, which have been incorporated to improve the manuscript.

Specific comments:

1. On the "Control" simulation: the model only accounted for the inter-annual variations of emission inventory for NOx and CO, but ignored that for NMVOCs. The authors argued that the US anthropogenic NMVOC emissions are much smaller than the natural ones and are hence negligible. The effect of the reduction in anthropogenic NMVOC emissions in US on the ozone trends should be evaluated, especially in the urban areas. Furthermore, the changes of the NMVOC emissions in other regions (ca. Europe and Asia) should be also taken into account, as it may influence the subsequent modeling estimation of the Asian contribution to the US ozone trend. At least, sensitivity modeling studies should be done to check if considering the changes of NMVOC emissions in different regions could affect the conclusions of the modeling study.

In our simulations, similar to the anthropogenic emissions for CO and NOx, global anthropogenic emissions of NMVOC use the REanalysis of the TROpospheric chemical composition (RETRO) monthly global inventory for 2000 (Hu et al., 2015). Emissions over China, rest of Asia, the US and Europe are further replaced by the MEIC (base year is 2008; www.meicmodel.org), INTEX-B (base year is 2006 (Zhang et al., 2009)), NEI05 (base year is 2005, ftp://aftp.fsl.noaa.gov/divisions/taq/), and EMEP (base year is 2005 (Auvray and Bey, 2005)) regional inventories, respectively.

To address this comment, we have conducted a sensitivity simulation, as discussed in the end of the revised Sect. 5.1:

"A sensitivity simulation was conducted to test the effect of anthropogenic NMVOC

emission changes not included in the "Control" simulation. In the sensitivity simulation, we scaled the NMVOC emissions to the years of 2004 and 2012 based on the EDGARv4.3.2 database which provides the emission time series (1970–2012). Emissions of NMVOC were scaled according to emissions over five regions (China, rest of Asia, the US, Europe and rest of world). Other emissions are the same with the "Control" simulation. In this sensitivity simulation, the modeled change in annual mean ozone from 2004 to 2012 is 1.7 ppb (equivalent to 0.21 ppb/yr) averaged over the US, with local ozone changes ranging from -1.9 to 8.1 ppb across the selected 124 grid cells. This magnitude of ozone change is consistent with the "Control" simulation results with the modeled ozone trend at 0.22 ppb/yr during 2004–2012 (Table 5). For the urban sites, the mean ozone change in the sensitivity simulation (2.1 ppb, or 0.26 ppb/yr) is also close to the "Control" simulation (0.28 ppb/yr). These results suggest that changes in anthropogenic NMVOC emissions have not led to a systemic ozone trend across the US on top of the effect of NOx emission changes, consistent with the results in Simon et al. (2015).".

2. Figure 3: the ozone growth rate during 2004-2012 is much faster than that in 1990-2014, especially during nighttime hours. The authors explain this in Section 5 partly due to the choice of beginning and end years. Is there any other reason for the stronger ozone trend in the recent decade?

We have added another possible reason in the second paragraph of the revised Sect. 5:

"A possible reason for the stronger ozone trend in the recent decade is that anthropogenic emissions of NOx decline much more rapidly (Fig. 4b) over 2004–2012 (at a rate of 4.1%/yr relative to 2004) than over 1990–2014 (2.1%/yr relative to 1990). Also, a heat wave swept much of the US in 2012, partly contributing to the high value in that year."

3. Section 3: it would be better if the authors can compare the observed ozone trends

in US with those from other regions of the world, such as Europe and Asia.

In the second paragraph of the revised Sect. 3, we have added the following information to compare the annual mean ozone trends across the globe:

"Similar to the enhanced US annual average ozone, increasing trends of ∼1 ppb/yr in the annual mean ozone are observed at mountainous sites (e.g., Tanimoto et al., 2009) and regional background stations (e.g., Wang et al., 2009) in Asia. In contrast, European annual mean ozone levels have on average been decreasing during the last 20 years (e.g., Sicard et al, 2013). Furthermore, annual mean surface ozone at a background station in eastern China has declined (Xu et al., 2008)."

In the third paragraph of the revised Sect. 3, we have also compared the summertime US ozone trends with those in Europe:

"Similarly, a substantial decrease in precursor concentrations over the last decades in Europe, in line with the long-term emission declines (Colette et al., 2011; Wilson et al., 2012), has resulted in a reduction in ozone episodes (Guerreiro et al., 2014). In contrast, the warm season afternoon ozone over eastern China has been growing at rates of 1–3 ppb/yr over the past 20 years (Sun et al., 2016; and references therein) as a result of rapidly growing precursor emissions."

4. Section 4: the analyses revealed the weaker correlations between climate variability and the nighttime ozone anomaly (compared to the daytime and daily average), and between climate variability and ozone anomalies over western US (compared to eastern US). The authors need comment on the possible reasons for the weak relationships between climate and nighttime ozone, and western US ozone.

In the fourth paragraph of the revised Sect. 4, we have added the possible reasons for the weak correlation between climate and nighttime ozone:

"A possible reason for weaker correlations between climate indices and the nighttime ozone anomaly (compared to the daytime and daily anomalies) is the distinct chemistry

at night (i.e., titration by nitrogen), compared to the daytime photochemistry. As the vertical mixing weakens at night, the chemical process becomes more localized and may be less sensitive to large-scale climate variability. "

In the third paragraph of the revised Sect. 4, we have added the possible reasons for the western US:

"This is likely due to compensating effects on different transport and chemical processes. For example, a positive AMONI are associated with enhanced ozone production (due to enhanced temperature and reduced precipitation in the context of a positive AMO), weakened trans-pacific ozone transport from Asia (due to a negative ONI with a La Niña pattern), and strengthened STE."

5. Page 12, Line 30-31: may the authors comment on the significance, quantitatively, of the Asian contribution in comparison with those from US and climate variability?

We have added the following information to the last paragraph in Sect. 5.2:

"The rising Asian emissions contribute to the US annual mean ozone trends (0.01–0.02 ppb/yr for daily mean, daytime and nighttime ozone) much less than the contributions from the US anthropogenic emission changes (0.08–0.22 ppb/yr) and climate variability (0.05–0.07 ppb/yr)."

This sentence is a summary of the detailed discussion on various contributing factors in the revised Sect. 5.2.

6. Tables 1-3: provide the unit, ppbv yr-1?

Yes, we have added the unit.

7. Table 4: what do the numbers in this table mean? Correlation coefficient (r or r2)?

It means the linear Pearson correlation coefficient (r). We have added this information in the table.

8. Figure 4a: provide p-values for the trends.

We have added the p-values in the revised Figure 4a.

9. Page 6, Line 32: delete one "mean".

Deleted.

Please also note the supplement to this comment:
https://www.atmos-chem-phys-discuss.net/acp-2017-659/acp-2017-659-AC1-supplement.pdf

**Supplement:**

**Referee #1**

The authors analyzed the inter-annual variability and trends of daytime, nighttime and daily mean ozone during 1990-2014 over the United States based on air quality monitoring data at about 1000 stations, and also assessed the impacts of anthropogenic emissions versus climate variability on the ozone trends during 2004-2012 by the GEOS-Chem modeling. This work combines observations and global modeling to evaluate the ozone trends and driving factors in the past two decades over US, both diurnally and spatially, and provide useful information about the ozone trends at the non-peak hours. The manuscript is clearly organized and well written, and the interpretation of the observational and modeling results is also fairly well. I recommend that this paper can be considered for publication after the following comments being addressed.

We thank the reviewer for comments, which have been incorporated to improve the manuscript.

Specific comments:

1. On the "Control" simulation: the model only accounted for the inter-annual variations of emission inventory for NOx and CO, but ignored that for NMVOCs. The authors argued that the US anthropogenic NMVOC emissions are much smaller than the natural ones and are hence negligible. The effect of the reduction in anthropogenic NMVOC emissions in US on the ozone trends should be evaluated, especially in the urban areas. Furthermore, the changes of the NMVOC emissions in other regions (ca. Europe and Asia) should be also taken into account, as it may influence the subsequent modeling estimation of the Asian contribution to the US ozone trend. At least, sensitivity modeling studies should be done to check if considering the changes of NMVOC emissions in different regions could affect the conclusions of the modeling study.

In our simulations, similar to the anthropogenic emissions for CO and NOx, global anthropogenic emissions of NMVOC use the REanalysis of the TROpospheric chemical composition (RETRO) monthly global inventory for 2000 (Hu et al., 2015). Emissions over China, rest of Asia, the US and Europe are further replaced by the MEIC (base year is 2008; www.meicmodel.org), INTEX-B (base year is 2006 (Zhang et al., 2009)), NEI05 (base year is 2005, ftp://aftp.fsl.noaa.gov/divisions/taq/), and EMEP (base year is 2005 (Auvray and Bey, 2005)) regional inventories, respectively.

To address this comment, we have conducted a sensitivity simulation, as discussed in the end of the revised Sect. 5.1:

"A sensitivity simulation was conducted to test the effect of anthropogenic NMVOC emission changes not included in the "Control" simulation. In the sensitivity simulation, we scaled the NMVOC emissions to the years of 2004 and 2012 based on the EDGARv4.3.2 database which provides the emission time series (1970–2012). Emissions of NMVOC were scaled according to emissions over five regions (China,

rest of Asia, the US, Europe and rest of world). Other emissions are the same with the "Control" simulation.

In this sensitivity simulation, the modeled change in annual mean ozone from 2004 to 2012 is 1.7 ppb (equivalent to 0.21 ppb/yr) averaged over the US, with local ozone changes ranging from -1.9 to 8.1 ppb across the selected 124 grid cells. This magnitude of ozone change is consistent with the "Control" simulation results with the modeled ozone trend at 0.22 ppb/yr during 2004–2012 (Table 5). For the urban sites, the mean ozone change in the sensitivity simulation (2.1 ppb, or 0.26 ppb/yr) is also close to the "Control" simulation (0.28 ppb/yr). These results suggest that changes in anthropogenic NMVOC emissions have not led to a systemic ozone trend across the US on top of the effect of NOx emission changes, consistent with the results in Simon et al. (2015).".

2. Figure 3: the ozone growth rate during 2004-2012 is much faster than that in 1990-2014, especially during nighttime hours. The authors explain this in Section 5 partly due to the choice of beginning and end years. Is there any other reason for the stronger ozone trend in the recent decade?

We have added another possible reason in the second paragraph of the revised Sect. 5:

"A possible reason for the stronger ozone trend in the recent decade is that anthropogenic emissions of $NO_x$ decline much more rapidly (Fig. 4b) over 2004–2012 (at a rate of 4.1%/yr relative to 2004) than over 1990–2014 (2.1%/yr relative to 1990). Also, a heat wave swept much of the US in 2012, partly contributing to the high value in that year."

3. Section 3: it would be better if the authors can compare the observed ozone trends in US with those from other regions of the world, such as Europe and Asia.

In the second paragraph of the revised Sect. 3, we have added the following information to compare the annual mean ozone trends across the globe:

"Similar to the enhanced US annual average ozone, increasing trends of ~1 ppb/yr in the annual mean ozone are observed at mountainous sites (e.g., Tanimoto et al., 2009) and regional background stations (e.g., Wang et al., 2009) in Asia. In contrast, European annual mean ozone levels have on average been decreasing during the last 20 years (e.g., Sicard et al, 2013). Furthermore, annual mean surface ozone at a background station in eastern China has declined (Xu et al., 2008)."

In the third paragraph of the revised Sect. 3, we have also compared the summertime US ozone trends with those in Europe:

"Similarly, a substantial decrease in precursor concentrations over the last decades in Europe, in line with the long-term emission declines (Colette et al., 2011; Wilson et al., 2012), has resulted in a reduction in ozone episodes (Guerreiro et al., 2014). In contrast, the warm season afternoon ozone over eastern China has been growing at

rates of 1–3 ppb/yr over the past 20 years (Sun et al., 2016; and references therein) as a result of rapidly growing precursor emissions."

4. Section 4: the analyses revealed the weaker correlations between climate variability and the nighttime ozone anomaly (compared to the daytime and daily average), and between climate variability and ozone anomalies over western US (compared to eastern US). The authors need comment on the possible reasons for the weak relationships between climate and nighttime ozone, and western US ozone.

In the fourth paragraph of the revised Sect. 4, we have added the possible reasons for the weak correlation between climate and nighttime ozone:

"A possible reason for weaker correlations between climate indices and the nighttime ozone anomaly (compared to the daytime and daily anomalies) is the distinct chemistry at night (i.e., titration by nitrogen), compared to the daytime photochemistry. As the vertical mixing weakens at night, the chemical process becomes more localized and may be less sensitive to large-scale climate variability. "

In the third paragraph of the revised Sect. 4, we have added the possible reasons for the western US:

"This is likely due to compensating effects on different transport and chemical processes. For example, a positive AMONI are associated with enhanced ozone production (due to enhanced temperature and reduced precipitation in the context of a positive AMO), weakened trans-pacific ozone transport from Asia (due to a negative ONI with a La Niña pattern), and strengthened STE."

5. Page 12, Line 30-31: may the authors comment on the significance, quantitatively, of the Asian contribution in comparison with those from US and climate variability?

We have added the following information to the last paragraph in Sect. 5.2:

"The rising Asian emissions contribute to the US annual mean ozone trends (0.01–0.02 ppb/yr for daily mean, daytime and nighttime ozone) much less than the contributions from the US anthropogenic emission changes (0.08–0.22 ppb/yr) and climate variability (0.05–0.07 ppb/yr)."

This sentence is a summary of the detailed discussion on various contributing factors in the revised Sect. 5.2.

6. Tables 1-3: provide the unit, ppbv yr-1?

Yes, we have added the unit.

7. Table 4: what do the numbers in this table mean? Correlation coefficient (r or r2)?

It means the linear Pearson correlation coefficient (r). We have added this information in the table.

8. Figure 4a: provide p-values for the trends.

We have added the p-values in the revised Figure 4a.

9. Page 6, Line 32: delete one "mean".

Deleted.

**Referee #2**

The authors have conducted a very interesting study to investigate the ozone trends at different times of a day. This work includes many important details and could have important implications for future ozone air quality control. So I recommend this paper could be published after they have addressed some issues as listed below.

We thank the reviewer for comments, which have been incorporated to the revised manuscript.

Major comments.

When trying to quantify the influence of meteorology on U.S. ozone trends during 1990-2014, the authors focused on two climate patterns—AMO and ONI. A new index named AMONI is also constructed. Even they have found strong correlations between the annual mean ozone with these climate indices, but the underlying physical mechanism is unclear to the readers. It may be too reckless to say US ozone is dependent on AMONI. Here are my concerns.

There are a lot of evidence in the literature on the relevance of AMO and ONI (ENSO) to the US ozone. We discussed some evidence in the original manuscript, and we have further improved the discussion in the revised Sect. 2.2:

[revised manuscript text omitted]

1. According to a new study (Shen et al., 2017), the U.S. summertime ozone is associated with a tripole SST pattern in the Atlantic and a dipole pattern in the northeastern Pacific. This raises the question that why the authors choose to use AMO and ONI. Is this the best choice?

See above for the choice of AMO and ONI.

The correlation between AMONI and ozone (0.71 for daytime mean ozone and 0.62 for daily mean ozone) is comparable to the correlation (~ 0.7) found by Shen et al. (2017).

We have added a discussion of Shen et al. (2017) in the revised Sect. 2.2.:

"A recent work by Shen et al. (2017) developed two metrics, MAM-ΔSST and MAM-ΔSLP, to study June-July-August (JJA) MDA8 ozone variability across much of the eastern US. They found that MAM-ΔSST is highly correlated to the summer ozone (R = ~ 0.7), which level of correlation is comparable to our results for daytime and daily mean ozone (see Sect. 4)."

2. The mechanism related to ENSO as proposed by Xu et al. [2017] only applies to the spring or fall season (e.g. the subtropical jet wind only explains the springtime ozone decreases in a small region in the southwest US). Since annual mean ozone is used in this study, the authors can't simply use the mechanism in Xu et al. [2017] to support their conclusion here.

See above for why we chose the ONI index.

Xu et al. (2017) showed that over 1993–2013, the monthly ozone decreases (increases) during El Niño (La Niña) years, with the amplitude varying from 0.4 ppb (for the US average) up to 1.8 ppb (for the southeastern US) per standard deviation of the Niño 3.4 index (our ONI index). Also, Fig. 1b in Xu et al. (2017) shows that the correlation coefficient between monthly ozone anomalies and the Niño 3.4 index are negative across the whole US. Our calculated correlation between US annual mean ozone anomalies and the ONI index are also negative (Figure 6c).

Xu et al. (2017) showed that the largest ENSO influences occur over two southern US regions during fall and over two western US regions during winter to spring. They showed that ENSO affects surface ozone via chemical processes during the warm seasons in the southern regions, where favorable meteorological conditions occur, but via dynamic transport during the cold seasons in western regions, where the ENSO-induced circulation variations are large. Our seasonal and regional AMONI-ozone correlation results (Table 4), with higher correlations during summer and fall over the eastern US regions and during winter and spring over the US western

regions, are also consistent with Xu et al. (2017).

We have added more discussion on the seasonal and regional features of AMONI-ozone correlations in the revised Sect. 4.

3. The influence of AMO on U.S. weather varies by season and region. See Sutton et al. [2007] for more details. Also, AMO not only influence the temperature but also the subsidence, precipitation, drought and surface wind. Throughout this paper, the authors only mention the impact on temperature, which is not enough.

We have added discussions of AMO-related and ozone-relevant climate impacts, including temperature, precipitation, droughts and atmospheric circulation, in the revised Sect. 2.2.

We have also updated the analysis to explain the AMONI-ozone correlations in the revised Sect. 4. In particular, we show that "A positive AMONI anomaly is associated with increased temperature over the east (Fig. 8(c)), which enhances biogenic emissions, changes chemical reaction rates, and changes atmospheric circulation that overall lead to increased ozone formation/buildup (Jacob and Winner, 2009; Shen et al., 2015; Fu et al., 2015; Xu et al., 2017).".

4. Does annual mean ozone strongly depend on temperature? It seems the authors mainly mention the influence of temperature changes when they explain the relationship of AMONI and annual mean ozone.

We have revised the third paragraph in Sect. 4:

"Figure 7(a, b) further shows the correlation between de-trended AMONI and de-trended daily and daytime mean ozone in individual grid cells. The AMONI-ozone correlation is positive and statistically significant over most of the eastern US, and it reaches 0.82 over parts of the southeast. A positive AMONI anomaly is associated with increased temperature over the east (Fig. 8(c)), which enhances biogenic emissions, changes chemical reaction rates, and changes atmospheric circulation that overall lead to increased ozone formation/buildup (Jacob and Winner, 2009; Shen et al., 2015; Fu et al., 2015; Xu et al., 2017). AMONI also correlate positively to ozone over the high-altitude west. This is because a negative ONI anomaly (La Niña-like) means a decrease in lower-tropospheric transport of ozone-poor air from the Eastern Pacific (Xu et al., 2017) and a more meandering subtropical jet and strengthened ozone transport from the stratosphere that compensates for weakened transport from Asia (Lin et al., 2015). AMONI correlates negatively to ozone over southern California, likely reflecting reduced temperature there associated with a positive AMONI (negative ONI) anomaly (Fig. 8(b, c))."

Minor comments.

P1 L32. Should mention that his mechanism only applies to the surface ozone.

We have changed this line to "Chemically, surface ozone is produced in the sunshining daytime and destroyed mainly by nitrogen oxides ($NO_x$) at night".

P1 L35. We usually use "MDA8" instead of "DM8A".

We have changed "DM8A" to "MDA8" throughout the text.

P2 L1-7. The summary here is too general. Try to give more details or you can just cite two review papers (Jacob and Winner, 2009; Fiore et al., 2015).

We have simplified this sentence to "Previous observational and modeling studies have revealed important impacts of varying climate conditions and anthropogenic precursor emissions on the near-surface daytime, MDA8 or daily mean ozone over the United States (US) (Jacob and Winner, 2009; Fiore et al., 2015)."

P2 L19-24. Move this paragraph to the last paragraph of the introduction part.

Moved and updated.

P2 L25-34. These studies choose to focus on one region or one season due to potential occurrence of ozone episodes. So why should we care about the low ozone part, e.g. the ozone in the wintertime?

The low-level ozone still affects health (Bell et al., 2006; Peng et al., 2013; Yang et al., 2012), as reviewed in the first paragraph of the introduction. In addition, analysis of their trends and underlying driving factors helps to understand the regional and seasonal ozone changes in the context of reductions in anthropogenic emissions and warming in the climate.

P3 L8-9. What is the meaning of "21.3-28.5% of data are missing"? It is unclear to the readers.

We have changed this sentence to "The fraction of hours in any year with missing data ranges from 21.3% to 28.5%."

P4 L13-15. Please make it clear why you choose these two climate indices?

Please see our response to major comment 1.

P4 L19-22. The summary of the influence of AMO on U.S. weather is too general.

Updated. Please see our response to major comment 1.

P4 L16-23. What definition of AMO is used here? What SST product (ERSST V4 or HadISST?) is used to calculate the AMO index? Does Fu 2015 mention that AMO can enhance the temperature anomaly and ozone production in US? They just speculate that AMO could change the ozone transport between the north and south US. Also to my knowledge, the influence of AMO on US weather varies in different seasons. Please refer to Sutton et al. [2007].

Updated. Please see our response to major comment 1.

Fu et al. (2015) does not mention that AMO could enhance the temperature anomaly and ozone production in US. And we did not imply this. We have modified the text for clarification.

P4 L24-30. What SST product is used here? Why the authors choose to use Nino 3.4 rather than other indices like Nino 1+2, Nino3 or Nino 4?

Please see our response to major comment 1.

P4 L31-36. Please specify the detrending method here. Is there a reason that you want to subtract the influence of ONI from AMO?

The detrended AMO and ONI indices were downloaded from the National Oceanic and Atmospheric Administration (NOAA) website. The method used for these two indices are linear detrending (Enfield et al., 2001).

The negative sign for ONI in the formula accounts for the negative correlation between de-trended ozone and ONI anomalies (see more discussion of AMONI in Sect. 2.2).

P8 L33. Change "Relation" to "Relationship".

Modified.

P8 L35-37. Why do you use linear detrending given the fact that the domestic NOx emissions are not linearly decreased?

Here we would like to show the relationship between interannual ozone variability and interannual climate variability. Linear de-trending allows to keep all signals of interannual fluctuations other than linear trend.

P9 L13-16. The evidence shown here is not sufficient to support that positive AMONI leads to enhanced temperatures. This may be just a coincidence. None of the three references listed here can support the conclusion here.

We did not intend to imply the causality between AMONI and temperature. Rather, we regarded AMONI as an indicator of large-scale climate variability that is also associated with temperature. We put in the revised Sect. 4 that "A positive AMONI anomaly is associated with increased temperature over the east (Fig. 8(c)), which enhances biogenic emissions, changes chemical reaction rates, and changes atmospheric circulation that overall lead to increased ozone formation/buildup (Jacob and Winner, 2009; Shen et al., 2015; Fu et al., 2015; Xu et al., 2017)."

P9 L17. Can the mechanism proposed in Xu et al. [2017] be able to explain the relationship? Xu's mechanism only applies to the springtime ozone decreases in a small region in the southwest US, while the authors here are discussing the annual mean ozone.

Please see our response to major comment 2.

P9 L26-29. It is good to try AO, PDO and NAO. But there are also a lot of other climate indices. This doesn't mean AMONI is the best. See the metrics used in Shen et al. [2017].

Please see our response to major comment 1. Further, we have discussed Shen et al.

(2017) in the revised Sect. 2.2.

P10 L16-24. The reason is that a heatwave swept across much of the US in 2012. The authors may need to mention this here.

Updated. Thanks for the suggestion.

P11 L9. Do you use the emissions in 2004 here?

Yes, we fixed global anthropogenic emissions at the 2004 levels here.

P12 L19-27. I appreciate the author's efforts in quantifying the contribution of anthropogenic emissions and meteorological variability. But are the influences of these two processes independent? Why R is used here? Is it better to use R2?

We expect the two mechanisms to be largely independent here, as typically assumed. We have redefined the contribution of anthropogenic emission as $C_{anth} = R_{anth}^2/(R_{anth}^2 + R_{clim}^2)$, which has led to stronger contrast between the contributions of emissions and climate. The revised Fig. 11 shows that the emission contribution dominates in the nighttime hours (relatively constant at about 70%), with a reduction in the morning hours, an increase in the late afternoon hours, and a minimum value (at 10%) around 15:00.

---

## Author Comment (AC2) · 23 Nov 2017

Referee #2

The authors have conducted a very interesting study to investigate the ozone trends at different times of a day. This work includes many important details and could have important implications for future ozone air quality control. So I recommend this paper could be published after they have addressed some issues as listed below.

We thank the reviewer for comments, which have been incorporated to the revised manuscript.

Major comments.

When trying to quantify the influence of meteorology on U.S. ozone trends during 1990-2014, the authors focused on two climate patterns—AMO and ONI. A new index named AMONI is also constructed. Even they have found strong correlations between the annual mean ozone with these climate indices, but the underlying physical mechanism is unclear to the readers. It may be too reckless to say US ozone is dependent on AMONI. Here are my concerns.

There are a lot of evidence in the literature on the relevance of AMO and ONI (ENSO) to the US ozone. We discussed some evidence in the original manuscript, and we have further improved the discussion in the revised Sect. 2.2:

"We relate the interannual variability of ozone to two major climate indices relevant to the US air quality (Sutton et al., 2007; Lin et al., 2015): the Atlantic Multi-decadal Oscillation (AMO; https://www.esrl.noaa.gov/psd/data/timeseries/AMO/) index and the Oceanic Niño index (ONI; http://origin.cpc.ncep.noaa.gov/products/analysis_monitoring/ensostuff/ONI_v5.php).

De-trended annual AMO index time series over 1990–2014 is calculated from the unsmoothed Kaplan sea surface temperature (SST) dataset of the National Oceanic and Atmospheric Administration (NOAA) Earth System Research Laboratory (http://www.esrl.noaa.gov/psd/data/correlation/amon.us.data). Observational and model studies have shown that the recent multi-decadal fluctuations in Atlantic SST were associated with large-scale climate anomalies (Enfield et al., 2001; Johannessen et al., 2004; Hu et al., 2011; Oglesby et al., 2012) important for the ozone chemistry. The warm conditions in the North Atlantic Ocean (positive AMO) have been associated with increased air temperatures at northern latitudes (by more than 3°C locally especially in winter) (Johannessen et al., 2004) and reduced summertime rainfall and increased droughts over much of the US (Enfield et al., 2001; McCabe et al., 2004). Delworth and Mann (2000) showed evidence of AMO-related variations in sea level pressure (SLP), and hence atmospheric circulation, in the North Atlantic region. Such AMO-associated meteorological changes can alter the distribution of tropospheric constituents (Olsen et al., 2016), including ozone (Shen et al., 2015; Lin et al., 2014). For example, warmer temperature under a positive AMO phase tends to enhance the ozone production over the US (Lin et al., 2014); less precipitation means less cloudy and higher radiation for photochemistry (Kunkel et al., 2008); and the change in circulation affects the ozone transport (Fu et al., 2015).

De-trended annual ONI index time series over 1990–2014 is calculated from the NOAA Climate Prediction Center dataset (http://www.cpc.noaa.gov/products/analysis_monitoring/ensostuff/ensoyears.shtml). The ONI index refers to ERSST.v4 SST anomalies in the Niño 3.4 region, as an indicator of the El Niño-Southern Oscillation (ENSO). These SST anomalies are associated with global climate variability, including changes in the temperature and precipitation patterns over the Unites States (Ropelewski and Halpert, 1987; Yu et al., 2012; Yu and Zou, 2013; Liang et al., 2015), through alterations in atmospheric circulations and teleconnection patterns (Bjerknes, 1969; Enfield, 1989). The Niño 3.4 index (referred as ONI here) has been widely used to examine the impact of ENSO on surface ozone over the United States (e.g., Lin et al., 2015; Xu et al., 2017). Xu et al. (2017) showed that over 1993–2013, the monthly ozone decreases (increases) during El Niño (La Niña) years, with the amplitude varying from 0.4 ppb (for the US average) up to 1.8 ppb (for the southeastern US) per standard deviation of the Niño 3.4 index. The La Niña years (strongly negative ONI) tend to be associated with a more meandering jet over the central western US in favor of stratosphere-troposphere exchange (STE) of ozone but not transport from Asia (Lin et al., 2015).

In order to indicate the climate variability that influence the whole US, we combined the de-trended and normalized AMO and ONI indices to obtain a third index, named AMONI: AMONI=ãĂŰAMOãĂŮ_(ãĂŰdetrendedãĂŮ_normalized )-ãĂŰONIãĂŮ_(ãĂŰdetrendedãĂŮ_normalized ). The normalization could adjust the AMO and ONI values measured on different scales to a common scale and keep the individual characteristics of the original AMO and ONI indices. The negative sign for

ONI in the formula accounts for the negative correlation between de-trended ozone and ONI anomalies (see Sect. 4). Thus a positive AMO and a negative ONI, both of which are closely related to higher ozone mixing ratios over the US, contribute to a positive AMONI index.

A recent work by Shen et al. (2017) developed two metrics, MAM-$\Delta$SST and MAM-$\Delta$SLP, to study June-July-August (JJA) MDA8 ozone variability across much of the eastern US. They found that MAM-$\Delta$SST is highly correlated to the summer ozone (R = $\sim$ 0.7), which level of correlation is comparable to our results for daytime and daily mean ozone (see Sect. 4)."

1. According to a new study (Shen et al., 2017), the U.S. summertime ozone is associated with a tripole SST pattern in the Atlantic and a dipole pattern in the northeastern Pacific. This raises the question that why the authors choose to use AMO and ONI. Is this the best choice?

See above for the choice of AMO and ONI.

The correlation between AMONI and ozone (0.71 for daytime mean ozone and 0.62 for daily mean ozone) is comparable to the correlation ($\sim$ 0.7) found by Shen et al. (2017).

We have added a discussion of Shen et al. (2017) in the revised Sect. 2.2.:

"A recent work by Shen et al. (2017) developed two metrics, MAM-$\Delta$SST and MAM-$\Delta$SLP, to study June-July-August (JJA) MDA8 ozone variability across much of the eastern US. They found that MAM-$\Delta$SST is highly correlated to the summer ozone (R = $\sim$ 0.7), which level of correlation is comparable to our results for daytime and daily mean ozone (see Sect. 4)."

2. The mechanism related to ENSO as proposed by Xu et al. [2017] only applies to the spring or fall season (e.g. the subtropical jet wind only explains the springtime ozone decreases in a small region in the southwest US). Since annual mean ozone is used in this study, the authors can't simply use the mechanism in Xu et al. [2017] to support

their conclusion here.

See above for why we chose the ONI index.

Xu et al. (2017) showed that over 1993–2013, the monthly ozone decreases (increases) during El Niño (La Niña) years, with the amplitude varying from 0.4 ppb (for the US average) up to 1.8 ppb (for the southeastern US) per standard deviation of the Niño 3.4 index (our ONI index). Also, Fig. 1b in Xu et al. (2017) shows that the correlation coefficient between monthly ozone anomalies and the Niño 3.4 index are negative across the whole US. Our calculated correlation between US annual mean ozone anomalies and the ONI index are also negative (Figure 6c).

Xu et al. (2017) showed that the largest ENSO influences occur over two southern US regions during fall and over two western US regions during winter to spring. They showed that ENSO affects surface ozone via chemical processes during the warm seasons in the southern regions, where favorable meteorological conditions occur, but via dynamic transport during the cold seasons in western regions, where the ENSO-induced circulation variations are large. Our seasonal and regional AMONI-ozone correlation results (Table 4), with higher correlations during summer and fall over the eastern US regions and during winter and spring over the US western regions, are also consistent with Xu et al. (2017).

We have added more discussion on the seasonal and regional features of AMONI-ozone correlations in the revised Sect. 4.

3. The influence of AMO on U.S. weather varies by season and region. See Sutton et al. [2007] for more details. Also, AMO not only influence the temperature but also the subsidence, precipitation, drought and surface wind. Throughout this paper, the authors only mention the impact on temperature, which is not enough.

We have added discussions of AMO-related and ozone-relevant climate impacts, including temperature, precipitation, droughts and atmospheric circulation, in the revised

[Figure]

Sect. 2.2.

We have also updated the analysis to explain the AMONI-ozone correlations in the revised Sect. 4. In particular, we show that "A positive AMONI anomaly is associated with increased temperature over the east (Fig. 8(c)), which enhances biogenic emissions, changes chemical reaction rates, and changes atmospheric circulation that overall lead to increased ozone formation/buildup (Jacob and Winner, 2009; Shen et al., 2015; Fu et al., 2015; Xu et al., 2017).".

4. Does annual mean ozone strongly depend on temperature? It seems the authors mainly mention the influence of temperature changes when they explain the relationship of AMONI and annual mean ozone.

We have revised the third paragraph in Sect. 4:

"Figure 7(a, b) further shows the correlation between de-trended AMONI and de-trended daily and daytime mean ozone in individual grid cells. The AMONI-ozone correlation is positive and statistically significant over most of the eastern US, and it reaches 0.82 over parts of the southeast. A positive AMONI anomaly is associated with increased temperature over the east (Fig. 8(c)), which enhances biogenic emissions, changes chemical reaction rates, and changes atmospheric circulation that overall lead to increased ozone formation/buildup (Jacob and Winner, 2009; Shen et al., 2015; Fu et al., 2015; Xu et al., 2017). AMONI also correlate positively to ozone over the high-altitude west. This is because a negative ONI anomaly (La Niña-like) means a decrease in lower-tropospheric transport of ozone-poor air from the Eastern Pacific (Xu et al., 2017) and a more meandering subtropical jet and strengthened ozone transport from the stratosphere that compensates for weakened transport from Asia (Lin et al., 2015). AMONI correlates negatively to ozone over southern California, likely reflecting reduced temperature there associated with a positive AMONI (negative ONI) anomaly (Fig. 8(b, c))."

Minor comments.

P1 L32. Should mention that his mechanism only applies to the surface ozone.

We have changed this line to "Chemically, surface ozone is produced in the sunshining daytime and destroyed mainly by nitrogen oxides (NOx) at night".

P1 L35. We usually use "MDA8" instead of "DM8A".

We have changed "DM8A" to "MDA8" throughout the text.

P2 L1-7. The summary here is too general. Try to give more details or you can just cite two review papers (Jacob and Winner, 2009; Fiore et al., 2015).

We have simplified this sentence to "Previous observational and modeling studies have revealed important impacts of varying climate conditions and anthropogenic precursor emissions on the near-surface daytime, MDA8 or daily mean ozone over the United States (US) (Jacob and Winner, 2009; Fiore et al., 2015)."

P2 L19-24. Move this paragraph to the last paragraph of the introduction part.

Moved and updated.

P2 L25-34. These studies choose to focus on one region or one season due to potential occurrence of ozone episodes. So why should we care about the low ozone part, e.g. the ozone in the wintertime?

The low-level ozone still affects health (Bell et al., 2006; Peng et al., 2013; Yang et al., 2012), as reviewed in the first paragraph of the introduction. In addition, analysis of their trends and underlying driving factors helps to understand the regional and seasonal ozone changes in the context of reductions in anthropogenic emissions and warming in the climate.

P3 L8-9. What is the meaning of "21.3-28.5% of data are missing"? It is unclear to the readers.

We have changed this sentence to "The fraction of hours in any year with missing data

ranges from 21.3% to 28.5%."

P4 L13-15. Please make it clear why you choose these two climate indices?

Please see our response to major comment 1.

P4 L19-22. The summary of the influence of AMO on U.S. weather is too general.

Updated. Please see our response to major comment 1.

P4 L16-23. What definition of AMO is used here? What SST product (ERSST V4 or HadISST?) is used to calculate the AMO index? Does Fu 2015 mention that AMO can enhance the temperature anomaly and ozone production in US? They just speculate that AMO could change the ozone transport between the north and south US. Also to my knowledge, the influence of AMO on US weather varies in different seasons. Please refer to Sutton et al. [2007].

Updated. Please see our response to major comment 1.

Fu et al. (2015) does not mention that AMO could enhance the temperature anomaly and ozone production in US. And we did not imply this. We have modified the text for clarification.

P4 L24-30. What SST product is used here? Why the authors choose to use Nino 3.4 rather than other indices like Nino 1+2, Nino3 or Nino 4?

Please see our response to major comment 1.

P4 L31-36. Please specify the detrending method here. Is there a reason that you want to subtract the influence of ONI from AMO?

The detrended AMO and ONI indices were downloaded from the National Oceanic and Atmospheric Administration (NOAA) website. The method used for these two indices are linear detrending (Enfield et al., 2001).

The negative sign for ONI in the formula accounts for the negative correlation between

de-trended ozone and ONI anomalies (see more discussion of AMONI in Sect. 2.2).

P8 L33. Change "Relation" to "Relationship".

Modified.

P8 L35-37. Why do you use linear detrending given the fact that the domestic NOx emissions are not linearly decreased?

Here we would like to show the relationship between interannual ozone variability and interannual climate variability. Linear de-trending allows to keep all signals of interannual fluctuations other than linear trend.

P9 L13-16. The evidence shown here is not sufficient to support that positive AMONI leads to enhanced temperatures. This may be just a coincidence. None of the three references listed here can support the conclusion here.

We did not intend to imply the causality between AMONI and temperature. Rather, we regarded AMONI as an indicator of large-scale climate variability that is also associated with temperature. We put in the revised Sect. 4 that "A positive AMONI anomaly is associated with increased temperature over the east (Fig. 8(c)), which enhances biogenic emissions, changes chemical reaction rates, and changes atmospheric circulation that overall lead to increased ozone formation/buildup (Jacob and Winner, 2009; Shen et al., 2015; Fu et al., 2015; Xu et al., 2017)."

P9 L17. Can the mechanism proposed in Xu et al. [2017] be able to explain the relationship? Xu's mechanism only applies to the springtime ozone decreases in a small region in the southwest US, while the authors here are discussing the annual mean ozone.

Please see our response to major comment 2.

P9 L26-29. It is good to try AO, PDO and NAO. But there are also a lot of other climate indices. This doesn't mean AMONI is the best. See the metrics used in Shen et al.

[Figure]

[2017].

Please see our response to major comment 1. Further, we have discussed Shen et al. (2017) in the revised Sect. 2.2.

P10 L16-24. The reason is that a heatwave swept across much of the US in 2012. The authors may need to mention this here.

Updated. Thanks for the suggestion.

P11 L9. Do you use the emissions in 2004 here?

Yes, we fixed global anthropogenic emissions at the 2004 levels here.

P12 L19-27. I appreciate the author's efforts in quantifying the contribution of anthropogenic emissions and meteorological variability. But are the influences of these two processes independent? Why R is used here? Is it better to use R2?

We expect the two mechanisms to be largely independent here, as typically assumed. We have redefined the contribution of anthropogenic emission as $C\_anth=(R\_anth^2)/((R\_anth^2+R\_clim^2))$, which has led to stronger contrast between the contributions of emissions and climate. The revised Fig. 11 shows that the emission contribution dominates in the nighttime hours (relatively constant at about 70%), with a reduction in the morning hours, an increase in the late afternoon hours, and a minimum value (at 10%) around 15:00.

Please also note the supplement to this comment:
https://www.atmos-chem-phys-discuss.net/acp-2017-659/acp-2017-659-AC2-supplement.pdf

[Figure]

**Supplement:**

**Referee #1**

The authors analyzed the inter-annual variability and trends of daytime, nighttime and daily mean ozone during 1990-2014 over the United States based on air quality monitoring data at about 1000 stations, and also assessed the impacts of anthropogenic emissions versus climate variability on the ozone trends during 2004-2012 by the GEOS-Chem modeling. This work combines observations and global modeling to evaluate the ozone trends and driving factors in the past two decades over US, both diurnally and spatially, and provide useful information about the ozone trends at the non-peak hours. The manuscript is clearly organized and well written, and the interpretation of the observational and modeling results is also fairly well. I recommend that this paper can be considered for publication after the following comments being addressed.

We thank the reviewer for comments, which have been incorporated to improve the manuscript.

Specific comments:

1. On the "Control" simulation: the model only accounted for the inter-annual variations of emission inventory for NOx and CO, but ignored that for NMVOCs. The authors argued that the US anthropogenic NMVOC emissions are much smaller than the natural ones and are hence negligible. The effect of the reduction in anthropogenic NMVOC emissions in US on the ozone trends should be evaluated, especially in the urban areas. Furthermore, the changes of the NMVOC emissions in other regions (ca. Europe and Asia) should be also taken into account, as it may influence the subsequent modeling estimation of the Asian contribution to the US ozone trend. At least, sensitivity modeling studies should be done to check if considering the changes of NMVOC emissions in different regions could affect the conclusions of the modeling study.

In our simulations, similar to the anthropogenic emissions for CO and NOx, global anthropogenic emissions of NMVOC use the REanalysis of the TROpospheric chemical composition (RETRO) monthly global inventory for 2000 (Hu et al., 2015). Emissions over China, rest of Asia, the US and Europe are further replaced by the MEIC (base year is 2008; www.meicmodel.org), INTEX-B (base year is 2006 (Zhang et al., 2009)), NEI05 (base year is 2005, ftp://aftp.fsl.noaa.gov/divisions/taq/), and EMEP (base year is 2005 (Auvray and Bey, 2005)) regional inventories, respectively.

To address this comment, we have conducted a sensitivity simulation, as discussed in the end of the revised Sect. 5.1:

"A sensitivity simulation was conducted to test the effect of anthropogenic NMVOC emission changes not included in the "Control" simulation. In the sensitivity simulation, we scaled the NMVOC emissions to the years of 2004 and 2012 based on the EDGARv4.3.2 database which provides the emission time series (1970–2012). Emissions of NMVOC were scaled according to emissions over five regions (China,

rest of Asia, the US, Europe and rest of world). Other emissions are the same with the "Control" simulation.

In this sensitivity simulation, the modeled change in annual mean ozone from 2004 to 2012 is 1.7 ppb (equivalent to 0.21 ppb/yr) averaged over the US, with local ozone changes ranging from -1.9 to 8.1 ppb across the selected 124 grid cells. This magnitude of ozone change is consistent with the "Control" simulation results with the modeled ozone trend at 0.22 ppb/yr during 2004–2012 (Table 5). For the urban sites, the mean ozone change in the sensitivity simulation (2.1 ppb, or 0.26 ppb/yr) is also close to the "Control" simulation (0.28 ppb/yr). These results suggest that changes in anthropogenic NMVOC emissions have not led to a systemic ozone trend across the US on top of the effect of NOx emission changes, consistent with the results in Simon et al. (2015).".

2. Figure 3: the ozone growth rate during 2004-2012 is much faster than that in 1990-2014, especially during nighttime hours. The authors explain this in Section 5 partly due to the choice of beginning and end years. Is there any other reason for the stronger ozone trend in the recent decade?

We have added another possible reason in the second paragraph of the revised Sect. 5:

"A possible reason for the stronger ozone trend in the recent decade is that anthropogenic emissions of $NO_x$ decline much more rapidly (Fig. 4b) over 2004–2012 (at a rate of 4.1%/yr relative to 2004) than over 1990–2014 (2.1%/yr relative to 1990). Also, a heat wave swept much of the US in 2012, partly contributing to the high value in that year."

3. Section 3: it would be better if the authors can compare the observed ozone trends in US with those from other regions of the world, such as Europe and Asia.

In the second paragraph of the revised Sect. 3, we have added the following information to compare the annual mean ozone trends across the globe:

"Similar to the enhanced US annual average ozone, increasing trends of ~1 ppb/yr in the annual mean ozone are observed at mountainous sites (e.g., Tanimoto et al., 2009) and regional background stations (e.g., Wang et al., 2009) in Asia. In contrast, European annual mean ozone levels have on average been decreasing during the last 20 years (e.g., Sicard et al, 2013). Furthermore, annual mean surface ozone at a background station in eastern China has declined (Xu et al., 2008)."

In the third paragraph of the revised Sect. 3, we have also compared the summertime US ozone trends with those in Europe:

"Similarly, a substantial decrease in precursor concentrations over the last decades in Europe, in line with the long-term emission declines (Colette et al., 2011; Wilson et al., 2012), has resulted in a reduction in ozone episodes (Guerreiro et al., 2014). In contrast, the warm season afternoon ozone over eastern China has been growing at

rates of 1–3 ppb/yr over the past 20 years (Sun et al., 2016; and references therein) as a result of rapidly growing precursor emissions."

4. Section 4: the analyses revealed the weaker correlations between climate variability and the nighttime ozone anomaly (compared to the daytime and daily average), and between climate variability and ozone anomalies over western US (compared to eastern US). The authors need comment on the possible reasons for the weak relationships between climate and nighttime ozone, and western US ozone.

In the fourth paragraph of the revised Sect. 4, we have added the possible reasons for the weak correlation between climate and nighttime ozone:

"A possible reason for weaker correlations between climate indices and the nighttime ozone anomaly (compared to the daytime and daily anomalies) is the distinct chemistry at night (i.e., titration by nitrogen), compared to the daytime photochemistry. As the vertical mixing weakens at night, the chemical process becomes more localized and may be less sensitive to large-scale climate variability. "

In the third paragraph of the revised Sect. 4, we have added the possible reasons for the western US:

"This is likely due to compensating effects on different transport and chemical processes. For example, a positive AMONI are associated with enhanced ozone production (due to enhanced temperature and reduced precipitation in the context of a positive AMO), weakened trans-pacific ozone transport from Asia (due to a negative ONI with a La Niña pattern), and strengthened STE."

5. Page 12, Line 30-31: may the authors comment on the significance, quantitatively, of the Asian contribution in comparison with those from US and climate variability?

We have added the following information to the last paragraph in Sect. 5.2:

"The rising Asian emissions contribute to the US annual mean ozone trends (0.01–0.02 ppb/yr for daily mean, daytime and nighttime ozone) much less than the contributions from the US anthropogenic emission changes (0.08–0.22 ppb/yr) and climate variability (0.05–0.07 ppb/yr)."

This sentence is a summary of the detailed discussion on various contributing factors in the revised Sect. 5.2.

6. Tables 1-3: provide the unit, ppbv yr-1?

Yes, we have added the unit.

7. Table 4: what do the numbers in this table mean? Correlation coefficient (r or r2)?

It means the linear Pearson correlation coefficient (r). We have added this information in the table.

8. Figure 4a: provide p-values for the trends.

We have added the p-values in the revised Figure 4a.

9. Page 6, Line 32: delete one "mean".

Deleted.

**Referee #2**

The authors have conducted a very interesting study to investigate the ozone trends at different times of a day. This work includes many important details and could have important implications for future ozone air quality control. So I recommend this paper could be published after they have addressed some issues as listed below.

We thank the reviewer for comments, which have been incorporated to the revised manuscript.

Major comments.

When trying to quantify the influence of meteorology on U.S. ozone trends during 1990-2014, the authors focused on two climate patterns—AMO and ONI. A new index named AMONI is also constructed. Even they have found strong correlations between the annual mean ozone with these climate indices, but the underlying physical mechanism is unclear to the readers. It may be too reckless to say US ozone is dependent on AMONI. Here are my concerns.

There are a lot of evidence in the literature on the relevance of AMO and ONI (ENSO) to the US ozone. We discussed some evidence in the original manuscript, and we have further improved the discussion in the revised Sect. 2.2:

[revised manuscript text omitted]

1. According to a new study (Shen et al., 2017), the U.S. summertime ozone is associated with a tripole SST pattern in the Atlantic and a dipole pattern in the northeastern Pacific. This raises the question that why the authors choose to use AMO and ONI. Is this the best choice?

See above for the choice of AMO and ONI.

The correlation between AMONI and ozone (0.71 for daytime mean ozone and 0.62 for daily mean ozone) is comparable to the correlation (~ 0.7) found by Shen et al. (2017).

We have added a discussion of Shen et al. (2017) in the revised Sect. 2.2.:

"A recent work by Shen et al. (2017) developed two metrics, MAM-ΔSST and MAM-ΔSLP, to study June-July-August (JJA) MDA8 ozone variability across much of the eastern US. They found that MAM-ΔSST is highly correlated to the summer ozone (R = ~ 0.7), which level of correlation is comparable to our results for daytime and daily mean ozone (see Sect. 4)."

2. The mechanism related to ENSO as proposed by Xu et al. [2017] only applies to the spring or fall season (e.g. the subtropical jet wind only explains the springtime ozone decreases in a small region in the southwest US). Since annual mean ozone is used in this study, the authors can't simply use the mechanism in Xu et al. [2017] to support their conclusion here.

See above for why we chose the ONI index.

Xu et al. (2017) showed that over 1993–2013, the monthly ozone decreases (increases) during El Niño (La Niña) years, with the amplitude varying from 0.4 ppb (for the US average) up to 1.8 ppb (for the southeastern US) per standard deviation of the Niño 3.4 index (our ONI index). Also, Fig. 1b in Xu et al. (2017) shows that the correlation coefficient between monthly ozone anomalies and the Niño 3.4 index are negative across the whole US. Our calculated correlation between US annual mean ozone anomalies and the ONI index are also negative (Figure 6c).

Xu et al. (2017) showed that the largest ENSO influences occur over two southern US regions during fall and over two western US regions during winter to spring. They showed that ENSO affects surface ozone via chemical processes during the warm seasons in the southern regions, where favorable meteorological conditions occur, but via dynamic transport during the cold seasons in western regions, where the ENSO-induced circulation variations are large. Our seasonal and regional AMONI-ozone correlation results (Table 4), with higher correlations during summer and fall over the eastern US regions and during winter and spring over the US western

regions, are also consistent with Xu et al. (2017).

We have added more discussion on the seasonal and regional features of AMONI-ozone correlations in the revised Sect. 4.

3. The influence of AMO on U.S. weather varies by season and region. See Sutton et al. [2007] for more details. Also, AMO not only influence the temperature but also the subsidence, precipitation, drought and surface wind. Throughout this paper, the authors only mention the impact on temperature, which is not enough.

We have added discussions of AMO-related and ozone-relevant climate impacts, including temperature, precipitation, droughts and atmospheric circulation, in the revised Sect. 2.2.

We have also updated the analysis to explain the AMONI-ozone correlations in the revised Sect. 4. In particular, we show that "A positive AMONI anomaly is associated with increased temperature over the east (Fig. 8(c)), which enhances biogenic emissions, changes chemical reaction rates, and changes atmospheric circulation that overall lead to increased ozone formation/buildup (Jacob and Winner, 2009; Shen et al., 2015; Fu et al., 2015; Xu et al., 2017).".

4. Does annual mean ozone strongly depend on temperature? It seems the authors mainly mention the influence of temperature changes when they explain the relationship of AMONI and annual mean ozone.

We have revised the third paragraph in Sect. 4:

"Figure 7(a, b) further shows the correlation between de-trended AMONI and de-trended daily and daytime mean ozone in individual grid cells. The AMONI-ozone correlation is positive and statistically significant over most of the eastern US, and it reaches 0.82 over parts of the southeast. A positive AMONI anomaly is associated with increased temperature over the east (Fig. 8(c)), which enhances biogenic emissions, changes chemical reaction rates, and changes atmospheric circulation that overall lead to increased ozone formation/buildup (Jacob and Winner, 2009; Shen et al., 2015; Fu et al., 2015; Xu et al., 2017). AMONI also correlate positively to ozone over the high-altitude west. This is because a negative ONI anomaly (La Niña-like) means a decrease in lower-tropospheric transport of ozone-poor air from the Eastern Pacific (Xu et al., 2017) and a more meandering subtropical jet and strengthened ozone transport from the stratosphere that compensates for weakened transport from Asia (Lin et al., 2015). AMONI correlates negatively to ozone over southern California, likely reflecting reduced temperature there associated with a positive AMONI (negative ONI) anomaly (Fig. 8(b, c))."

Minor comments.

P1 L32. Should mention that his mechanism only applies to the surface ozone.

We have changed this line to "Chemically, surface ozone is produced in the sunshining daytime and destroyed mainly by nitrogen oxides ($NO_x$) at night".

P1 L35. We usually use "MDA8" instead of "DM8A".

We have changed "DM8A" to "MDA8" throughout the text.

P2 L1-7. The summary here is too general. Try to give more details or you can just cite two review papers (Jacob and Winner, 2009; Fiore et al., 2015).

We have simplified this sentence to "Previous observational and modeling studies have revealed important impacts of varying climate conditions and anthropogenic precursor emissions on the near-surface daytime, MDA8 or daily mean ozone over the United States (US) (Jacob and Winner, 2009; Fiore et al., 2015)."

P2 L19-24. Move this paragraph to the last paragraph of the introduction part.

Moved and updated.

P2 L25-34. These studies choose to focus on one region or one season due to potential occurrence of ozone episodes. So why should we care about the low ozone part, e.g. the ozone in the wintertime?

The low-level ozone still affects health (Bell et al., 2006; Peng et al., 2013; Yang et al., 2012), as reviewed in the first paragraph of the introduction. In addition, analysis of their trends and underlying driving factors helps to understand the regional and seasonal ozone changes in the context of reductions in anthropogenic emissions and warming in the climate.

P3 L8-9. What is the meaning of "21.3-28.5% of data are missing"? It is unclear to the readers.

We have changed this sentence to "The fraction of hours in any year with missing data ranges from 21.3% to 28.5%."

P4 L13-15. Please make it clear why you choose these two climate indices?

Please see our response to major comment 1.

P4 L19-22. The summary of the influence of AMO on U.S. weather is too general.

Updated. Please see our response to major comment 1.

P4 L16-23. What definition of AMO is used here? What SST product (ERSST V4 or HadISST?) is used to calculate the AMO index? Does Fu 2015 mention that AMO can enhance the temperature anomaly and ozone production in US? They just speculate that AMO could change the ozone transport between the north and south US. Also to my knowledge, the influence of AMO on US weather varies in different seasons. Please refer to Sutton et al. [2007].

Updated. Please see our response to major comment 1.

Fu et al. (2015) does not mention that AMO could enhance the temperature anomaly and ozone production in US. And we did not imply this. We have modified the text for clarification.

P4 L24-30. What SST product is used here? Why the authors choose to use Nino 3.4 rather than other indices like Nino 1+2, Nino3 or Nino 4?

Please see our response to major comment 1.

P4 L31-36. Please specify the detrending method here. Is there a reason that you want to subtract the influence of ONI from AMO?

The detrended AMO and ONI indices were downloaded from the National Oceanic and Atmospheric Administration (NOAA) website. The method used for these two indices are linear detrending (Enfield et al., 2001).

The negative sign for ONI in the formula accounts for the negative correlation between de-trended ozone and ONI anomalies (see more discussion of AMONI in Sect. 2.2).

P8 L33. Change "Relation" to "Relationship".

Modified.

P8 L35-37. Why do you use linear detrending given the fact that the domestic NOx emissions are not linearly decreased?

Here we would like to show the relationship between interannual ozone variability and interannual climate variability. Linear de-trending allows to keep all signals of interannual fluctuations other than linear trend.

P9 L13-16. The evidence shown here is not sufficient to support that positive AMONI leads to enhanced temperatures. This may be just a coincidence. None of the three references listed here can support the conclusion here.

We did not intend to imply the causality between AMONI and temperature. Rather, we regarded AMONI as an indicator of large-scale climate variability that is also associated with temperature. We put in the revised Sect. 4 that "A positive AMONI anomaly is associated with increased temperature over the east (Fig. 8(c)), which enhances biogenic emissions, changes chemical reaction rates, and changes atmospheric circulation that overall lead to increased ozone formation/buildup (Jacob and Winner, 2009; Shen et al., 2015; Fu et al., 2015; Xu et al., 2017)."

P9 L17. Can the mechanism proposed in Xu et al. [2017] be able to explain the relationship? Xu's mechanism only applies to the springtime ozone decreases in a small region in the southwest US, while the authors here are discussing the annual mean ozone.

Please see our response to major comment 2.

P9 L26-29. It is good to try AO, PDO and NAO. But there are also a lot of other climate indices. This doesn't mean AMONI is the best. See the metrics used in Shen et al. [2017].

Please see our response to major comment 1. Further, we have discussed Shen et al.

(2017) in the revised Sect. 2.2.

P10 L16-24. The reason is that a heatwave swept across much of the US in 2012. The authors may need to mention this here.

Updated. Thanks for the suggestion.

P11 L9. Do you use the emissions in 2004 here?

Yes, we fixed global anthropogenic emissions at the 2004 levels here.

P12 L19-27. I appreciate the author's efforts in quantifying the contribution of anthropogenic emissions and meteorological variability. But are the influences of these two processes independent? Why R is used here? Is it better to use R2?

We expect the two mechanisms to be largely independent here, as typically assumed. We have redefined the contribution of anthropogenic emission as $C_{anth} = R_{anth}^2/(R_{anth}^2 + R_{clim}^2)$, which has led to stronger contrast between the contributions of emissions and climate. The revised Fig. 11 shows that the emission contribution dominates in the nighttime hours (relatively constant at about 70%), with a reduction in the morning hours, an increase in the late afternoon hours, and a minimum value (at 10%) around 15:00.